# Sample Efficient Demonstration Selection for In-Context Learning

**Kiran Purohit** [* 1]   **Venktesh V** [* 2]   **Sourangshu Bhattacharya** [1]   **Avishek Anand** [2]

## Abstract

The in-context learning paradigm with LLMs has been instrumental in advancing a wide range of natural language processing tasks. The selection of few-shot examples (exemplars / demonstration samples) is essential for constructing effective prompts under context-length budget constraints. In this paper, we formulate the exemplar selection task as a top-$m$ best arms identification problem. A key challenge in this setup is the exponentially large number of arms that need to be evaluated to identify the $m$-best arms. We propose CASE (**C**hallenger **A**rm **S**ampling for **E**xemplar selection), a novel sample-efficient *selective exploration* strategy that maintains a shortlist of "challenger" arms, which are current candidates for the top-$m$ arms. In each iteration, only one of the arms from this shortlist or the current top-$m$ set is pulled, thereby reducing sample complexity and, consequently, the number of LLM evaluations. Furthermore, we model the scores of exemplar subsets (arms) using a parameterized linear scoring function, leading to *stochastic linear bandits* setting. CASE achieves remarkable efficiency gains of up to **7**× speedup in runtime while requiring **7**× fewer LLM calls (87% reduction) without sacrificing performance compared to state-of-the-art exemplar selection methods. We release our **code and data.** [1]

## 1. Introduction

In-context learning (ICL) and Chain-of-Thought (COT) have emerged as important techniques for enhancing the capabilities of large language models (LLMs) across a range of natural language tasks (Yang et al., 2018; Roy & Anand, 2022; Nanekhan et al., 2025; V et al., 2023). ICL allows

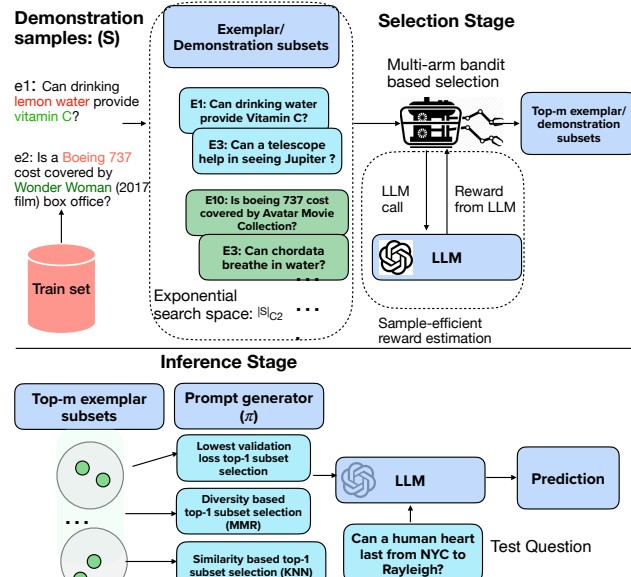

Figure 1: **Overview of CASE for selection of top-$m$ best exemplar subsets (arms).**

LLMs to perform tasks by conditioning on a context that includes demonstration examples or instructions, without the need for additional fine-tuning, making it flexible and adaptable. COT facilitates stepwise problem-solving by employing rationales. However, one key challenge in maximizing the effectiveness of ICL is the careful selection of few-shot examples along with their corresponding rationales (Lu et al., 2022; Zhao et al., 2021). We refer to the triplet of *(input, rationale, output)* as an exemplar/demonstration examples.

Most existing approaches for selecting demonstration examples are based on heuristics or trial-and-error methods (Fu et al., 2023; Brown et al., 2020), with only a few attempting to address the problem in a more principled way (Xiong et al., 2024). Exemplar / demonstration example selection can be classified into two categories: *task-level*, where a static set of exemplars representative of the task is chosen for inference, and *instance-level*, where exemplars are dynamically selected for each test instance during inference (Rubin et al., 2022; Xiong et al., 2024; Ye et al., 2023b), which introduces overhead during inference. Additionally, these methods do not consider the positive and negative

---
[*]Equal contribution. [1]Indian Institute of Technology, Kharagpur, India [2]Delft University of Technology (TU Delft), Netherlands.

*Proceedings of the $42^{nd}$ International Conference on Machine Learning*, Vancouver, Canada. PMLR 267, 2025. Copyright 2025 by the author(s).

[1]https://github.com/kiranpurohit/CASE

interactions between the exemplars. In contrast, selecting a static set of exemplars not only eliminates inference-time overhead but also enables prompt caching, allowing for the reuse of key-value (KV) attention states (Gim et al., 2024). In this work, we propose a hybrid approach where first multiple sets of exemplars are selected at task-level, and then test example-specific prompts are constructed at inference time over the reduced search space (task-level exemplars). The approaches closest to us are LENS (Li & Qiu, 2023), which use marginal gains in accuracy as selection metric, and EXPLORA (Purohit et al., 2024), which uses a heuristic contrastive learning based exploration scheme, incurring relatively high computational costs.

In this work, our primary goal is to propose a principled and sample efficient exemplar selection approach without compromising task performance. Recent studies aimed at developing theoretically grounded models for ICL (Zhang et al., 2023) primarily focus on linear functions and linear transformers. We propose to use a linear function based on sentence similarities between the in-context examples and validation examples as the surrogate model for modeling the goodness of in-context learning. This approach leads to a linear reward model for a multi-armed bandit-based exemplar selection, leading to the linear stochastic bandits setting (Abbasi-Yadkori et al., 2011). In this setting, our problem is posed a multiple exemplar-subset selection (MESS).

We formulate the selection of top-$m$ exemplar subsets as the problem of identification of top-$m$ arms (Réda et al., 2021) in the stochastic linear bandit setting (Réda et al., 2021). The GIFA framework (Réda et al., 2021) proposed an efficient gap-index based top-$m$ arms identification algorithm with reduced sample complexity. However, the computation of *challenger arm* (current candidates for top-$m$ arms), requires computation of gap-indices between all currently estimated top-$m$ arms and the remaining arms. This is impractical in our setting since each arm corresponds to a $k$-sized subset of the training exemplar set, leading to an exponential number of candidate arms. Hence we need an algorithm that can sample arms from the candidate sets. While some uniform sampling algorithms for top-$m$ arms identification exist, (Chen et al., 2017; Kaufmann & Kalyanakrishnan, 2013) for the general multi-arm bandit setting, to the best of our knowledge, there are no sampling-based algorithms for identification of top-$m$ arms in the stochastic linear bandits setting.

In this work, we propose CASE (**C**hallenger **A**rm **S**ampling for **E**xemplar Selection) where we propose a principled sampling of challenger arms to form a shortlist challenger set, pruning the space of possible candidate arms (see fig. 1). Our key idea is to iteratively create a low-regret set of selected challenger arms, in addition to the current top-$m$ arms, from uniformly sampled arms. This leads to a se-

lective exploration-based algorithm that is *sample-efficient*. We concurrently apply the state-of-the-art gap-index-based algorithm rule for selecting top-$m$ arms out of the total exploration set. We also provide theoretical arguments to justify our novel approach of combined selective exploration and gap-index based identification of top-$m$ arms. When applied to exemplar selection, we observe improvements in task performance of upto **15.19%** compared to state-of-the-art methods like LENS, competitive performance with competitive approaches like EXPLORA and reduces number of LLM calls (about **7x**) with speedup of upto **7x** when compared to state-of-the-art exemplar selection approaches.

## 2. Related Work

**Exemplar Selection for ICL.** The rise of LLMs has transformed them into general-purpose answering engines through emergent capabilities like ICL (Brown et al., 2020; Wei et al., 2022; 2023; Wang et al., 2023a; Kojima et al., 2023; Chen et al., 2022a) where a few examples are provided to LLMs to demonstrate the task. To eliminate manual selection, several automated methods have emerged, such as reinforcement learning (Zhang et al., 2022; Lu et al., 2023), trained retrievers (Xiong et al., 2024), Determinantal Point Processes (Ye et al., 2023a) and constrained optimization (Tonglet et al., 2023). Additionally, dynamic selection methods that are learning-free, such as similarity-based (Rubin et al., 2022), complexity-based (Fu et al., 2023), and MMR (Ye et al., 2023b), have been explored. However, dynamic methods increase inference-time computational costs. To address this, a pre-selected, representative set of exemplars is chosen for ICL, akin to coreset selection methods (Guo et al., 2022), though the key difference is that ICL does not involve parameter updates. To the best of our knowledge, there has been very little research on a sample efficient approach for exemplar selection, with the closest work being LENS (Li & Qiu, 2023) and EXPLORA (Purohit et al., 2024). LENS is expensive in number of LLM calls and is unsuitable for black-box models. EXPLORA proposes a heuristic exploration approach which is computationally intensive in terms of number of LLM calls and hence not sample efficient. We propose a novel sample efficient exemplar selection approach for black-box and open LLMs.

**Identification of Top-$m$ Arms in Stochastic Linear Bandits.** The top-$m$ arms identification problem aims to estimate a subset of $m$ arms with the highest means. Various methods have been proposed in both fixed-confidence (Kalyanakrishnan et al., 2012) and fixed-budget settings (Bubeck et al., 2013). In this paper, we focus on the fixed-confidence setting, where the error probability in estimating the top-$m$ arms should be smaller than a predefined parameter $\delta \in (0, 1)$. Adaptive sampling algorithms like UGapE (Gabillon et al., 2012) and LUCB (Kalyanakrishnan et al.,

2012), along with uniform sampling methods (Kaufmann & Kalyanakrishnan, 2013; Chen et al., 2017), have been introduced for the fixed confidence setup, but they lack efficiency in terms of sample complexity. While efficient adaptive sampling methods for linear bandits, such as (Fiez et al., 2019), RAGE (Zhang et al., 2023), LTS (Jedra & Proutiere, 2020), PEPS (Li et al., 2023), LinGapE (Xu et al., 2017) and LinGame (Degenne et al., 2020), have been proposed, they primarily address best-arm identification ($m = 1$). To the best of our knowledge, GIFA (Réda et al., 2021) was the first unified framework for efficient top-$m$ arm identification with low sample complexity. However, algorithms implemented in GIFA framework require large number of gap-index computations and comparisons, leading to high sample complexity. This is due to its challenger arm sampling mechanism, which considers the complement of the current top-$m$ estimate as the challenger set, resulting in a large search space. In this work, we propose a novel and efficient algorithm with applications to exemplar selection for diverse tasks in LLMs. Possible other formulations could be learning to rank (LTR) or online learning to rank (Zoghi et al., 2017; Grotov & De Rijke, 2016; Li et al., 2019). LTR relies on static relevance estimates assigned to samples or on user clicks, in an online setting. In contrast, our problem requires dynamic feedback from LLMs to determine the importance of exemplars. Therefore, learning to rank approaches are not well-suited to our problem setup.

## 3. CASE: Challenger Arm Sampling-based Exploration for Exemplar-subset Selection

In-context learning (ICL) enables LLMs to acquire task-specific knowledge from just a few demonstration examples without updating the model's parameter. However, due to the financial and computational costs and the limitations of LLMs in effectively processing large input contexts, providing all available demonstration examples for a given task is ineffective and impractical. Therefore, a key challenge is the *efficient and optimal selection of demonstration examples* (also called exemplars), particularly in a black-box setting where access to the model's parameters or confidence estimates is unavailable. To address this, we propose a novel algorithm for exemplar subset selection. We formulate the task of constructing an optimal prompt as a *multiple exemplar subset selection* problem (section 3.1). This is posed as a top-$m$ arm selection problem in stochastic linear bandits (Kalyanakrishnan & Stone, 2010; Réda et al., 2021) (section 3.2), leading to a novel sample efficient approach for top-$m$ arm selection, which we call as CASE (section 3.3).

### 3.1. Problem Setup- Multiple Exemplar-subset Selection

For a given task, let $u$ denote the input examples and $v$ denote the output labels. For simplicity, we include ratio-

nales / chain-of-thought reasoning used in the demonstration samples in the example $u$. Let $\mathcal{X} = \{u_i, v_i\}_{i=1}^n$ be the set of all $n$ potential exemplars and $(u_{test}, v_{test})$ denote a test input example and desired output. A prompt $P$ is constructed from a subset $S \subseteq \mathcal{X}$ of $k$ exemplars, which is used for the prediction of $u_{test}$. Hence, $P = [S, u_{test}] = [(u_{i_1}, v_{i_1}), ..., (u_{i_k}, v_{i_k}), u_{test}]$. The prompt $P$ is then passed to a response generator function $f$ which uses a sampling / decoding mechanism to generate responses from an LLM (given by the distribution $\mathbb{P}_{LLM}$. Hence, $f(P) \sim (\mathbb{P}_{LLM}(r|P))$. The final step is a post-processing $\delta$ applied to an LLM-generated response $f(P)$ for extracting the task-specific output $\hat{v}_{test}$. Commonly used post-processing strategies include regular-expression matching ($\delta_{regex}$) and self-consistency ($\delta_{SC}$) (Wang et al., 2023b).

While the above-described basic mechanism can be used for in-context learning, a single subset of $k$-exemplars often may not capture all the diverse aspects of a given task. Hence, prompt generators $\pi$ are used to generate prompts specific to a given test input $u_{test}$, from a shortlist of multiple high-scoring exemplar subsets $S_1, ..., S_m$, where $S_i \subseteq \mathcal{X}, \forall i = 1, ..., m$. Hence, prompt $P$ is calculated as $P = \pi(S_1, ..., S_m, u_{test})$. Popular prompt generators include similarity-based ($\pi_{KNN}$), which selects the subset with the highest semantic similarity to the test example, and diversity-based ($\pi_{MMR}$) (discussed in section 4). Hence, the entire output generation process can be described as:

$$\hat{v}_{test} = \delta(f(P)), \text{ where } P = \pi(U, u_{test}),$$
$$\text{and } U = (S_1, ..., S_m) \quad (1)$$

Here, $U = (S_1, ..., S_m)$ is a set of $m$ subsets of the potential exemplar set $\mathcal{X}$.

The main challenge in this paper is to select $U$, a high-scoring set of subsets of exemplars, $S_i$, that can be used by a given prompt generator $\pi$ to generate dynamic prompts (test example-specific) that maximize the performance on a given task. We use a separate validation set for measuring the performance of $U$, a selected set of exemplar subsets. Let $\mathcal{V}$ be the set of $n'$ validation examples $\{u'_i, v'_i\}_{i=1}^{n'}$, and let the validation accuracy for a prompt $P$ be defined as: $A(P, \mathcal{V}) = \frac{1}{n'} \sum_{i=1}^{n'} \mathbb{1}(v'_i = \delta(f(\pi(U, u'_i))))$. Our problem can be formulated as finding a set $U^*$ of $m$-subsets of $\mathcal{X}$ such that the corresponding prompt $P$ generated by the prompt generator $\pi$ maximizes the total validation accuracy.

$$U^* = \underset{U \in \mathcal{S}(\mathcal{X})^m}{\arg\max} A(\pi(U), \mathcal{V})$$

where $\mathcal{S}(\mathcal{X})$ is the set of all $k$-sized subsets of $\mathcal{X}$ \quad (2)

We call this as *multiple exemplar-subset selection* (MESS) formulation for finding the optimal prompt.

### 3.2. Top-$m$ Arm Selection formulation for MESS

The MESS problem defined above cannot be solved effectively using naive or heuristic search-based algorithms due to two reasons: (1) the search space $\mathcal{S}(\mathcal{X})^m$ is doubly exponentially large in $k$ and $m$ ($|\mathcal{S}(\mathcal{X})| = O(|\mathcal{X}|^k)$), and (2) the function $A(P, \mathcal{V})$ is computationally expensive due to LLM inference. In this section, we take a multi-armed bandit-based approach to implicitly estimate the function $A(P, \mathcal{V})$ in a sample-efficient manner, minimizing the number of queries to the LLM.

Let $a \in \{0, 1\}^n$ such that $\|a\|_0 = k$ denote the 1-hot encoding for an exemplar subset of size $k$, chosen from the potential exemplar set $\mathcal{X}$ ($|\mathcal{X}| = n$). The multi-armed bandit instance is constructed as arms $a_i \in \mathcal{S}(\mathcal{X})$ where $i \in \{1, ..., |\mathcal{S}(\mathcal{X})|\}$. Note that the number of arms is exponential in both $k$ and $n$. The reward for an arm $a$ is given by the accuracy $A(\pi(a), \mathcal{V})$, where $\pi$ is the prompt generated by the subset corresponding to $a$, and $\mathcal{V}$ is the validation set.

We further assume that the reward for an arm $a$, $\rho(a)$, can be modeled as a linear function of its features, aligning with the linear stochastic bandit setting (Abbasi-Yadkori et al., 2011). Intuitively, given an arm $a$, the *reward scoring function $\rho(a)$* should be a function of the similarity between the inputs of the selected exemplars, $u_i$, such that $a(i) = 1$, and the validation exemplar $u'_j \in \mathcal{V}$. In this work, we use a normalized BERT-based similarity score, $\text{Sim}_{ij} = \frac{\phi(u_i)^T \phi(u'_j)}{\|\phi(u_i)\|\|\phi(u'_j)\|}$, where $\phi(u)$ is the sentence encoding vector obtained from a pre-trained transformer model, e.g. SentenceBERT. Let, $\alpha = \{\alpha_i | i = 1, ..., n\}$ denote the vector of linear coefficients, where each $\alpha_i$ corresponds to the $i^{th}$ potential exemplar coefficient in the reward scoring function. Our linear model for the reward of an arm $a$ is given by:

$$\rho(a) \equiv \rho(a; \alpha) = \frac{1}{n'} \sum_{j=1}^{n'} \sum_{i=1}^{n} (\alpha_i a(i) \text{Sim}_{ij}) = \alpha^T x_a \quad (3)$$

where $x_a(i) = a(i) \left( \frac{1}{n'} \sum_{j=1}^{n'} \text{Sim}_{i,j} \right)$. Here, $x_a \in \mathbb{R}^n$ denotes the vector of features of the arm $a$, with non-zero components only for the exemplars that are part of the arm $a$. The observed reward is given by $\hat{\rho}(a; \alpha) = \rho(a; \alpha) + \eta$, where $\eta$ is a subgaussian noise, i.e. $\mathbb{E}[e^{\lambda \eta}] \leq \exp(\lambda^2 \xi^2 / 2)$, for some variance $\xi^2$. The MAB learning algorithm iteratively estimates the unknown coefficients ($\hat{\alpha}_t$), such that the empirical estimate of reward $\hat{\rho}_t(a) \equiv \hat{\rho}(a; \alpha_t) \approx \rho(a; \alpha)$ for the high scoring arms $a$ and a large time index $t > \tau$. Under the above multi armed bandit setup, we approximate the problem of MESS as finding the set of top-$m$ arms, denoted as $U^* = \{a_1, ..., a_m\}$.

Under this stochastic linear bandits assumption, the problem of identifying top-$m$ arms can be solved using the Gap-index based algorithms (Xu et al., 2018; Réda et al., 2021), which were unified under the GIFA framework (Réda et al., 2021). The GIFA framework is a class of iterative algorithms that maintain a set of estimated $m$-best arms, $U_t$. In each iteration $t$, the most ambiguous arm from $U_t$, say $b_t = \arg\max_{b \in U_t} \max_{a \in U_t^c} B_t(a, b)$, and the most ambiguous arm from $U_t^c$ (called the *challenger* arm), say $c_t = \arg\max_{c \in U_t^c} B_t(c, b_t)$, are computed. The arm with the highest variance between $b_t$ and $c_t$ is pulled, and the model parameters are updated. Here $B_t(a, b)$ is called the gap index between arms $a$ and $b$. For parameter updation, the design matrix $\hat{V}_{t+1}$ is computed as: $\hat{V}_{t+1} = \lambda I_n + \sum_{a \in \mathcal{S}} N_a x_a x_a^T$, where $N_a$ is the number of times an arm $a$ was pulled. The updated parameters $\hat{\alpha}_{t+1}$ are computed as: $\hat{\alpha}_{t+1} = (\hat{V}_{t+1})^{-1} (\sum_{l=1}^{t+1} r_l x_{a_l})$, where $r_l$ is the reward received by computing the accuracy of the prompt generated by the current arm. The gap-index between any two arms $i, j$ is computed as: $B_t(i, j) = \hat{\rho}_t(i) - \hat{\rho}_t(j) + W_t(i, j)$, where the confidence term is defined as: $W_t(i, j) = C_{t,\delta}(\|x_i\|_{\hat{\Sigma}_t^\lambda} + \|x_j\|_{\hat{\Sigma}_t^\lambda})$, where,

$$C_{t,\delta} = \sqrt{2 \ln \left( \frac{1}{\delta} \right) + N \ln \left( 1 + \frac{(t+1)L^2}{\lambda^2 N} \right)} + \frac{\sqrt{\lambda}}{\sigma} S, \; S \text{ and } L$$

are constants, $N$ is the total number of arms pulled, and $\hat{\Sigma}_t^\lambda = \sigma^2 (\hat{V}_t)^{-1}$.

### 3.3. CASE: Challenger-arm sampling-based top-$m$ arm selection

Implementing gap-index-based schemes for top-$m$ arm identification, for settings with exponentially large number of arms is infeasible. The key problem is to identify the most ambiguous arms in each iteration. We propose to mitigate this problem using: (a) identify a low-regret subset $N_t$ of $m'$ *next-best arms* after $U_t$, and (b) use a GIFA-based algorithm to identify the top-$m$ arms from the set $U_t \cup N_t$. The problem of combinatorial blowup of arms in the linear bandit setting has been sparsely studied, with selective exploration as one of the strategies (e.g. Algorithm 13, Chapter 23 of (Lattimore & Szepesvári, 2020)). Since it is impractical to explore all arms, the selective exploration scheme uniformly samples arms from the unexplored set and then selects the highest-scoring arms according to the current model. It then pulls the selected arms to update the model parameters.

Algorithm 1 describes the proposed *challenger-arm sampling* based exploration technique, called CASE. The set of top-m subsets (arms), $U_0$, is initialized to a random set sampled from $\mathcal{S}$. In lines $11 - 13$, we compute the updated $U_t$ by moving the highest scoring arm from $N_{t-1}$ if its score is higher than that of the lowest scoring arm in $U_{t-1}$, which is then moved to $N_{t-1}$. Using the selective exploration idea, CASE uniformly samples $m'$ arms from $(U_t \cup N_{t-1})^c$, to generate the set $M_t$, and then selects the top-$m'$ arms from $M_t \cup N_{t-1}$ to generate the updated $N_t$. $N_t \cup U_t$ is the high-reward selected set, from which

we explore using the selection rule. We use the greedy selection rule proposed in (Réda et al., 2021), where we select the arm that minimizes the variance between $s_t$ and $b_t$: $a^* = \arg\min_{a \in N_t \cup U_t} \|x_{b_t} - x_{s_t}\|_{(\hat{V}_{t-1} + x_a x_a^T)^{-1}}$. We call the LLM (line 21 in Algorithm 1) after the selection rule, where an arm (i.e., a set of exemplars) is sampled. Specifically, $A(\pi(a(t)), \mathcal{V})$ denotes the computation of accuracy on the validation subset, which requires LLM to generate predictions. This ensures that the LLM's rationale and output generation processes are explicitly integrated into our algorithm. By doing so, we effectively leverage the gap-index-based multi-arm bandit framework to optimize exemplar selection for ICL using LLMs. Steps 17 and 18 in algorithm 1 compute the revised most ambiguous arms $b_t \in U_t$ and $s_t \in N_t$. $s_t$ is *sampled challenger* arm selected from the selected set of next best arms $N_t$ using the gap index $B_t(a, b) = \hat{\rho}_t(a) - \hat{\rho}_t(b) + W_t(a, b)$. Finally, the revised parameters $\hat{\alpha}_{t+1}$ are computed using the revised design matrix $\hat{V}_{t+1}$ and the least squares estimation formulae described in lines 22 and 23 of algorithm 1. Note that, $\hat{V}_{t+1}$ can be computed from $\hat{V}_t$ using the incremental update formula $\hat{V}_{t+1} = \hat{V}_t + x_{a_{t+1}} x_{a_{t+1}}^T$. Hence, $(\hat{V}_{t+1})^{-1}$ can be calculated in $O(n^2)$ time using the Sherman-Morrison formula. We stop the updates when the convergence criteria for switching the arms between $U_t$ and $N_t$ have been achieved, i.e. $B_t(s_t, b_t) \leq \epsilon$. The time complexity for each iteration of our algorithm is $O(mm' + n^2 + \text{LLM\_inference\_time})$ which is due to lines 17, 21 and 23. Next, we discuss some theoretical results related to our method.

### 3.4. Sample Complexity bounds for the top-$m$ selection

The ability of CASE to identify the top-$m$ arms and correctly estimate the model parameters $\alpha$ rests on two arguments. Firstly, the selective exploration strategy results in a low regret set of arms $U_t \cup N_t$. Specifically, we assume the average regret (total regret / #iterations) of the set $U_T \cup N_T$ to upper bounded by $\epsilon$. While we postpone a rigorous derivation of the regret bound for CASE to a later study, we justify our assumption by using the SETC Algorithm (Algo 13 in (Lattimore & Szepesvári, 2020)), which follows a similar uniform exploration and commitment strategy in the linear bandit setting. SETC has a regret bound of $O(d\sqrt{n \log(n)})$ where $n$ is the number of time steps. Hence, the algorithm will achieve an average regret of at most $\epsilon$ after at most $\exp(W(\frac{\epsilon^2}{C^2 d^2}))$ timesteps, where $W$ is Lambert's function, and $C$ is the constant for the regret bound.

Secondly, once a low regret $U_T \cup N_T$ set is achieved, the gap-index-based algorithm correctly identifies the top-$m$ arms from the set $U_T \cup N_T$. Following (Réda et al., 2021), we obtain a high probability $(1 - \delta)$ upper bound for the sample complexity of CASE on the event $\mathcal{E} \triangleq \bigcap_{t>0} \bigcap_{i,j \in [K]} \left( \rho_i - \rho_j \in [-B_t(j, i), B_t(i, j)] \right)$, Let $\mathcal{S}_m^\star$

---

**Algorithm 1** CASE

1: **Input:** $\mathcal{X}$: set of all training exemplars, $k$: prompt size, $\mathcal{S}$ : all $k$-subsets of $\mathcal{X}$, $a \in \mathcal{S}$: an arm or $k$-subset
2: **Define:**   $U_t$: set of currently estimated top-$m$ arms.
3:            $N_t$: set of currently estimated next best-$m'$ arms.
4:            $b_t$: the most ambiguous arm from $U_t$
5:            $s_t$: the most ambiguous sampled arm from $N_t$
6: **Initialize:** $U_0 \leftarrow$ set of random $m$ arms from $\mathcal{S}$, $t \leftarrow 1$,  $\vec{\alpha}_1 \leftarrow \mathcal{N}(0, 1)$
7: **while** $\neg (B_t(s_t, b_t) \leq \epsilon)$ **do**
8:     *Construct $U_t$ by replacing $n_t$ with potentially better arm $c_t$*
9:     $n_t = \arg\min_{a \in U_{t-1}} \hat{\rho}_t(a)$
10:     $c_t = \arg\max_{a \in N_{t-1}} \hat{\rho}_t(a)$
11:     **if** $\hat{\rho}_t(c_t) \geq \hat{\rho}_t(n_t)$ **then**
12:         $U_t, N_t \leftarrow \text{swap}(n_t, c_t)$ from $U_{t-1}, N_{t-1}$
13:     **end if**
14:     $M_t \leftarrow s' \sim_{m'} (U_t \cup N_{t-1})^c$ // *Randomly sample*
15:     $N_t \leftarrow \text{top}_{m'}(M_t \cup N_{t-1}; \hat{\rho}_{(t)})$ //*Update $N_t$ from $N_{t-1}$ & $M_t$*
16:     *Compute the revised most ambiguous arms for convergence*
17:     $b_{t+1} = \arg\max_{b \in U_t} \max_{a \in N_t} [B_t(a, b)]$
18:     $s_{t+1} = \arg\max_{s \in N_t} [B_t(s, b_{t+1})]$
19:     *Pull selected arm, receive reward, and update parameters*
20:     $a_{t+1} \leftarrow \text{selection\_rule}(U_t, N_t)$
21:     $r_{t+1} = A(\pi(a(t)), \mathcal{V})$  // *LLM inference call*
22:     $\hat{V}_{t+1} = \lambda I_n + \sum_{a \in \mathcal{S}} N_a x_a x_a^T$ //*$\lambda$ regularized design matrix*
23:     $\hat{\alpha}_{t+1} = (\hat{V}_{t+1})^{-1}(\sum_{l=1}^{t+1} r_l x_{a_l})$ // *Least-squares estimate*
24:     $t \leftarrow t + 1$
25: **end while**
26: **Output:** $U_T$: Set of $m$ arms which have the highest reward

---

be the true set of top-$m$ arms. We define the true gap of an arm $i$ as $\Delta(i) \triangleq \rho(i) - \rho(m + 1)$ if $i \in \mathcal{S}_m^\star$, $\rho(m) - \rho(i)$ otherwise ($\Delta(i) \geq 0$ for any $i \in [K]$).

**Theorem 1.** *For* CASE*, on event $\mathcal{E}$ on which the algorithm is $(\varepsilon, m, \delta)$-PAC, stopping time $\tau_\delta$ satisfies $\tau_\delta \leq \inf\{u \in \rho^{*+} : u > 1 + H^\varepsilon(\mu) C_{\delta,u}^2 + \mathcal{O}(K)\}$, where, for algorithm with the largest variance selection rule[2] :*

$$H^\varepsilon(\mu) \triangleq 4\sigma^2 \sum_{a \in [K]} \max\left(\varepsilon, \frac{\varepsilon + \Delta_a}{3}\right)^{-2},$$

Above theorem essentially adopts the result in Theorem 2 from (Réda et al., 2021) to the setting where we restrict ourselves to arms in $U_T \cup N_T$. It mentions that the $\epsilon$-optimal top-m arms from $U_T \cup N_T$ are present in $U_T$ with prob. $1 - \delta$, if $T > \tau_\delta$. $K$ is the size of $U_T \cup N_T$.

**Proof Overview**: The proof builds upon the proofs for classical Top-$m$ linear bandits, LinGapE (Xu et al., 2018) and LinGIFA (Réda et al., 2021) while additionally accounting for the challenger arms shortlist in $N_t$ proposed in this work. To prove it, one of the key components is the following lemma, which holds for any gap indices of the form $B_t(i, j) \triangleq \hat{\rho}_t(i) - \hat{\rho}_t(j) + W_t(i, j)$ for $i, j \in [K]^2$.

---

[2]or pulling both arms in $\{b_t, c_t\}$ at time $t$

**Lemma 1.** *On the event $\mathcal{E}$, for all $t > 0$,*

$$B_t(s_t, b_t)(t) \leq \min(-(\Delta(b_t) \vee \Delta(s_t))$$
$$+2W_t(b_t, s_t), 0) + W_t(b_t, s_t)$$

*, where $a \vee b = max(a, b)$.*

Proof for Lemma 1 is provided in Appendix A.1. In summary, Theorem 1 and Lemma 1 together provide an upper bound on the expected number of arm pulls required by the algorithm, which translates to the number of LLM calls needed when applying CASE to complex reasoning tasks.

## 4. Experiments and Results

We aim to address the following research questions:
**RQ I.** How sample efficient is CASE compared to state-of-the-art exemplar selection and stochastic linear bandit methods?
**RQ II.** Can CASE, choose task-level demonstration samples without sacrificing task performance, compared to state-of-the-art exemplar selection methods?
**RQ III.** Can CASE work without exploration?

### 4.1. Experimental Setup

**Synthetic Experiments** For synthetic experiments, we adopt a setup similar to (Réda et al., 2021) and present results on simulated data. We set $\sigma = 0.5$, $\epsilon = 0$, and $\delta = 0.05$ across all experiments. Each experiment is conducted over 500 simulations. We explore various numbers of arms $K \in [7, 10, 20]$, and set the number of top arms to be identified with the highest means as $m = 3$. We choose a challenger set $N_t$ of size 3 and feature dimension is 3.

Task level Exemplar Selection for ICL

**Datasets and Metrics:** We evaluate on well-known datasets that require numerical and commonsense reasoning. For numerical reasoning, we use GSM8K, FinQA, TabMWP and AquaRAT; for commonsense reasoning, we use StrategyQA. Detailed descriptions of the datasets are provided in Appendix A.3. We report performance using the official metrics: Exact Match (EM) and Cover-EM (Rosset et al., 2021) for respective datasets.

**Hyperparameters (LLMs):** For LLMs, we set the temperature to 0.3 to reduce randomness. To reduce repetition, we apply a presence penalty of 0.6 and a frequency penalty of 0.8. The $max\_length$ for generation is set to 900.

**Hyperparameters (CASE):** We begin by clustering the training set into 5 clusters. We then form the set of exemplar subsets ($\mathcal{S}$), by sampling one exemplar from each cluster *with replacement*. This approach allows us to explore various combinations of exemplars. Note that the training set to form $\mathcal{S}$ are same across baselines for a fair comparison.

We set $m = |U_t| = 10$ to identify the top scoring subsets (arms) and choose a challenger set $N_t$ of size 5. We set the number of validation examples $\mathcal{V}$ to, 20 and $\epsilon = 0.1$.

**Baselines:** We compare against instance-level exemplar selection methods, such as MMR (Ye et al., 2023b) and KNN (Rubin et al., 2022), which use diversity and similarity-based measures to select exemplars for each test example. We configure MMR with $\lambda = 0.5$ to balance similarity and diversity. We also evaluate against coreset selection methods like Graphcut and Facility Location (Iyer & Bilmes, 2013). Finally, we compare with LENS (Li & Qiu, 2023) and EXPLORA (Purohit et al., 2024), state-of-the-art task-level exemplar selection approaches.

**CASE Hybrid Variants:** We also propose hybrid variants of CASE, where we select instance-level exemplars subset from the top-$m$ subsets identified by CASE using KNN or MMR, thereby reducing the search space. Our hybrid approach scores entire subsets by aggregating the similarity scores of each exemplar within the subset to the test instance, thereby preserving the interactions between exemplars.

### 4.2. Performance of CASE Compared to Existing Gap-Index-Based Approaches

To address **RQ1**, we conduct synthetic experiments following the setup in Section 4.1. We evaluate metrics such as average runtime, the average number of comparisons for gap index computation, for multi-armed bandit (MAB) approaches including LinGapE, LinGIFA, and CASE. The results for $K = 20$, $m = 3$, and $N = 3$ are shown in Figure 2. Other synthetic experiments are presented in Appendix A.9. In Figure 2(a), we observe that CASE significantly reduces the number of comparisons required for gap index computations compared to state-of-the-art gap-index-based MAB approaches like LinGapE, LinGIFA. This improvement is largely due to the principled challenger arm sampling strategy, pruning the space of possible challenger arms ($N_t$), resulting in efficient gap index computations. The efficiency gains stem from the fact that $|N_t| < |U_t^c|$, where existing gap-index frameworks use the entire $U_t^c$ to select challenger arms. From Figure 2(b), we also observe that runtime of CASE is lower when compared to LinGapE (about **5.6x**) and LinGIFA (about **12x**) due to lower number of arm comparisons and gap-index computations. In Figures 2(c) and 2(d), we analyze the gap index ($B_t(s_t, b_t)$) and simple regret across rounds (averaged across simulations) and observe that they approach 0 as the number of rounds increases. This demonstrates that, CASE samples good arms with our shortlist of $N_t$ serving as a good approximation of challenger arms. CASE converges with much lower gap index computations and has lower runtime compared to existing state-of-the-art gap-index MAB algorithms.

Since, EXPLORA uses an exploration based setting for se-

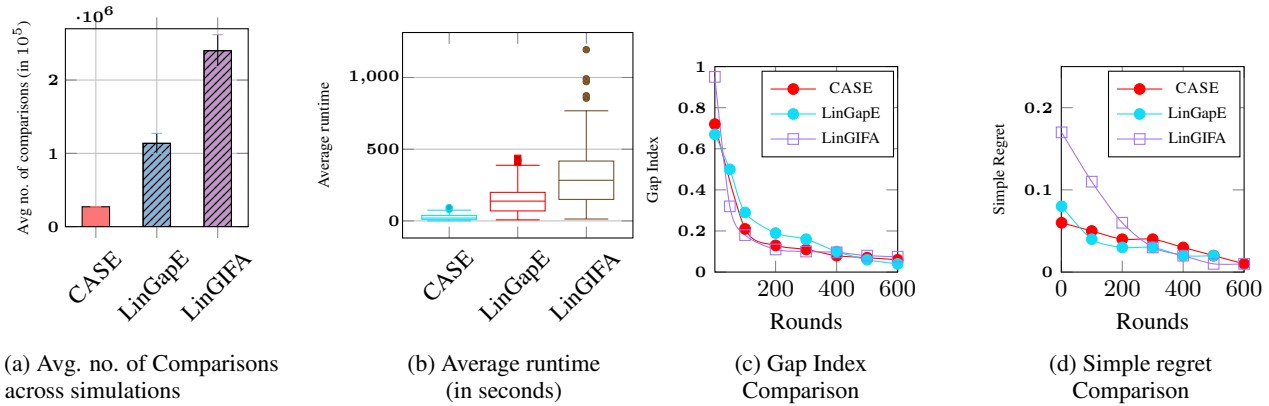

(a) Avg. no. of Comparisons across simulations
(b) Average runtime (in seconds)
(c) Gap Index Comparison
(d) Simple regret Comparison

Figure 2: Top-$m$ arm identification by CASE, LinGIFA and LinGapE for K=20, m=3, N=3.

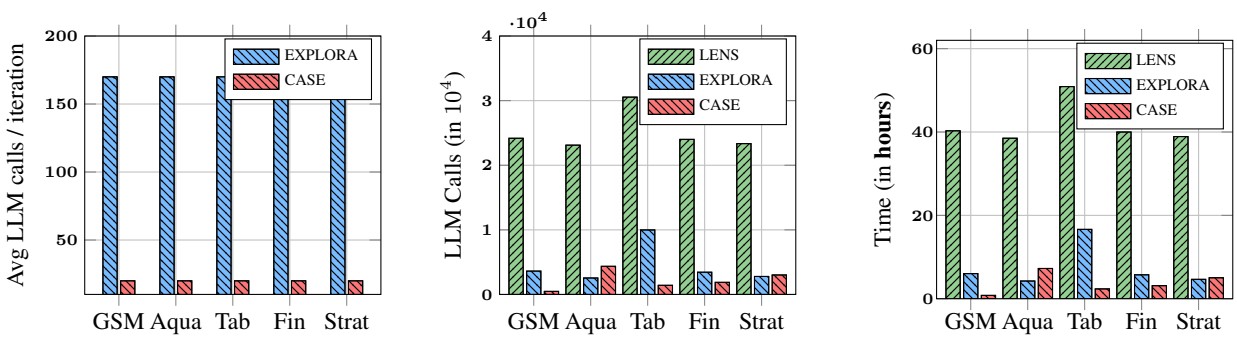

(a) Average number of LLM calls/iteration.
(b) LLM calls LENS, EXPLORA vs CASE
(c) Runtime LENS, EXPLORA vs CASE

Figure 3: Sample efficiency of CASE compared to LENS and EXPLORA.

lecting ICL demonstrations, we compare number of LLM calls per-iteration as shown in Figure 3a. We observe that CASE requires **8** × less calls per-iteration on average per iteration, proving to be extremely sample efficient. We also compare the total number of LLM calls made by CASE, EXPLORA and LENS (shown in Figure 3(b)) for the application of exemplar selection for ICL. We observe that CASE reduces LLM calls significantly compared to LENS (about **6x to 10.5x**). CASE is also sample efficient when compared to EXPLORA requiring upto **7x** less LLM calls (reduces number of LLM calls from 9986 when using EX-PLORA for TabMWP to 1420 in CASE & 3619 to 480 for GSM8K ) and also reduces the exemplar selection time by **7x** (shown in Figure 3(c)). We observe that the total number of LLM calls for CASE is slightly higher than EX-PLORA on AquaRat. This is primarily because EXPLORA is heuristic and stops after 10 iterations, while CASE runs till convergence criteria is met. However, as mentioned earlier, per-iteration CASE requires **8x** fewer calls than EXPLORA rendering it more sample efficient. This efficiency gains are due to extensive LLM calls by LENS for all training samples. EXPLORA though more efficient than LENS, employs a heuristic approach for arm pulls (LLM

calls with exemplar subsets) which results in exploration of less significant exemplar subsets (arms) and is less sample efficient. In contrast, CASE employs a novel challenger set sampling mechanism dramatically reduces the number of subsets (arms) that need to be explored and evaluated.

## 4.3. Performance Comparison on Exemplar Selection

To address **RQ II**, we compare CASE and its variants against state-of-the-art exemplar selection methods. Table 1 shows that CASE consistently outperforms random, as well as Few-Shot COT (Wei et al., 2023) methods, which rely on random or hand-picked samples without accounting for the interactions between exemplars and task-level performance. We observe that smaller LLMs are unable to fit more than 5 exemplars due to context length limits, and their performance plateaus beyond this point (Table 2). Hence, we employ 5-shot examples in our approach and all baselines. Furthermore, classical coreset selection methods like Graph Cut and Facility Location, perform worse or on par with random selection as they were not designed for the ICL paradigm. We also observe that CASE outperforms dynamic selection methods like KNN, PromptPG, and MMR. We also report the performance of CASE on open-source

Table 1: Results across datasets (we use 5-shot for all methods). Percentage improvements are reported over EXPLORA (Purohit et al., 2024). † indicates statistical significance (t-test) over EXPLORA at 0.05 level and ‡ at 0.01 level over LENS. Additionally, all CASE and it's variants are statistically significant over LENS at 0.05 level and not indicated here.

| Method | GSM8K | AquaRat | TabMWP | FinQA | StrategyQA |
|---|---|---|---|---|---|
| | | **GPT-3.5-turbo** | | | |
| **Instance level** | | | | | |
| KNN (S-BERT) (Rubin et al., 2022) | 53.07 | 52.75 | 77.95 | 52.65 | 81.83 |
| MMR (Ye et al., 2023b) | 54.36 | 51.18 | 77.32 | 49.87 | 82.86 |
| KNN+SC (Wang et al., 2023b) | 80.21 | 62.59 | 83.08 | 54.49 | 83.88 |
| MMR+SC (Wang et al., 2023b) | 78.01 | 59.45 | 81.36 | 50.74 | 83.88 |
| PromptPG (Lu et al., 2023) | - | - | 68.23 | 53.56 | - |
| **Task level** | | | | | |
| Manual Few-Shot COT (Wei et al., 2023) | 73.46 | 44.88 | 71.22 | 52.22 | 73.06 |
| Random | 67.79 | 49.80 | 55.89 | 53.70 | 81.02 |
| PS+ (Wang et al., 2023a) | 59.30 | 46.00 | - | - | - |
| GraphCut (Iyer & Bilmes, 2013) | 66.19 | 47.24 | 60.45 | 52.31 | 80.00 |
| FacilityLocation (Iyer & Bilmes, 2013) | 68.61 | 48.43 | 67.66 | 36.79 | 81.63 |
| Active-prompt (Diao et al., 2024) | 71.20 | 51.57 | - | - | 74.49 |
| EXPLORA (Purohit et al., 2024) | 77.86 | 53.54 | 83.07 | 59.46 | 85.71 |
| LENS (Li & Qiu, 2023) | 69.37 | 48.82 | 77.27 | 54.75 | 79.79 |
| **Our Approach** | | | | | |
| **CASE** | 79.91(▲2.63%) ‡ | 54.72(▲2.20%) | 83.42(▲0.04%)‡ | 59.72(▲0.43%) | 84.49 |
| **Hybrid Variants (Ours)** | | | | | |
| **CASE+KNN+SC** | **87.49**(▲12.36%)†‡ | **64.17**(▲19.85%)†‡ | **86.23**(▲3.80%)‡ | **64.25** (▲8.05%)†‡ | 85.92‡ |
| **CASE+MMR+SC** | 85.60 (▲9.94%)†‡ | 62.60(▲16.92%)†‡ | 85.91(▲3.41%)‡ | 63.47(▲6.74%)†‡ | 84.69‡ |
| | | **GPT-4o-mini** | | | |
| LENS (Li & Qiu, 2023) | 76.19 | 64.56 | 86.34 | 69.31 | 92.85 |
| **CASE** | **91.13** | **73.23** | **89.73** | **72.89** | **95.92** |

models like Mistral-7b and Llama2-7b (see **Appendix A.4**). Beyond the efficiency gained during inference, task-level exemplars demonstrate greater robustness (see **Appendix** A.5) compared to dynamic methods. The independent selection of exemplars in dynamic methods may result in negative interactions or skill redundancy, where lexically different exemplars may represent similar skills. To further support the claim that CASE selects more representative task-level exemplars, we conduct a *qualitative analysis* comparing exemplars chosen by LENS and CASE (see **Appendix A.8**). We find that CASE consistently selects exemplars with diverse skills required for solving tasks similar to EXPLORA, while being significantly efficient, whereas LENS selects exemplars with redundant skills. Hence, we observe that CASE is *competitive* with EXPLORA in terms of task performance while being significantly **sample-efficient** (upto **7x**) which is the primary goal of our work. For CASE, we evaluate the performance of gpt-3.5-turbo by **re-using exemplars** selected by smaller models, such as Llama2-7b (see **Appendix A.6**, Figure 4). In Table 1 we show the results

Table 2: Different values of k for Manual k-shot COT.

| Datasets | GSM | Aqua | Tab | Fin | Strat |
|---|---|---|---|---|---|
| Zero-shot | 67.02 | 38.15 | 57.10 | 47.51 | 59.75 |
| 1-shot | 67.55 | 38.58 | 66.30 | 49.26 | 68.16 |
| 3-shot | 68.99 | 41.33 | 70.50 | 51.93 | 70.00 |
| 5-shot | **73.46** | **44.88** | **71.22** | 52.22 | **73.06** |
| 7-shot | 68.84 | 44.88 | 70.09 | 52.26 | 70.61 |

of exemplar reuse from Mistral-7b. We observe that exemplars selected by CASE using smaller LLMs transfer well to larger LLMs, promoting exemplar reuse and efficiency.

### 4.4. Ablation Studies

To answer **RQ3**, we conduct several ablation studies (shown in Table 3), including one-time sampling of all exemplar subsets and a variant of CASE without exploration, to highlight the need for an exploration-based MAB approach. In one-time sampling approach, we sample each subset once, evaluate them on validation set, and select the subset with lowest validation loss. Inference results using the selected subset are presented in Table 3. We observe that one-time sampling underperforms compared to CASE as it evaluates all arms only once and lacks sufficient information to confidently identify the best arms. This method tends to overfit the validation set and leads to suboptimal selection without incorporating exploration or exploitation mechanisms.

Table 3: Ablation studies: one-time sampling, w/o exploration vs proposed exploration (CASE).

| Datasets | GSM | Aqua | Tab | Fin | Strat |
|---|---|---|---|---|---|
| One-time sampling | 76.72 | 50.39 | 81.23 | 54.14 | 80.20 |
| CASE (-exploration) | 76.57 | 47.64 | 77.17 | 45.95 | 80.00 |
| **CASE ( from Llama)** | **77.79** | **56.30** | **83.65** | **57.72** | **82.24** |
| **CASE (from Mistral)** | **79.91** | **54.72** | **83.42** | **59.72** | **84.49** |

Furthermore, we conduct an ablation where we fit the linear model once on a set of $\mathcal{S}$ subsets (as shown in Table 3) and select the subset with the highest mean for test set inference. Unlike CASE, which models the top-$m$ subsets in each round, this ablation fits the linear model once for all subsets. Since, this ablation is devoid of exploration mechanism it does not result in confident estimation of top-m arms resulting in suboptimal performance.

## 5. Conclusion

In this work, we propose an efficient exemplar subset selection method that identifies highly informative exemplar subsets for ICL. Our approach significantly reduces the number of LLM calls, offering a clear advantage over state-of-the-art methods in terms of sample efficiency without sacrificing task performance. Additionally, exemplars selected using smaller LLMs can be reused for larger models. In the future, we also plan to extend and apply CASE to adaptive retrieval methods scrnarios (Rathee et al., 2025b;a;c) and question answering (Venktesh et al., 2025). Specifically, using the challenger arm sampling techniques we could carefully choose relevant documents to re-rank from the first stage retrieval given LLM signals to make robust prompts. This is particularly useful for question that need quantitative, numerical, and temporal matching (Wallat et al., 2025; Venktesh et al., 2024). We also plan to derive a rigorous regret bound for CASE, and the role of exemplars in ICL.

## Impact Statement

Our work focuses on *efficient exemplar selection* to enhance in-context learning, emphasizing both *efficiency* and *sustainability*. By improving sample efficiency, our approach reduces the number of calls to LLMs, thereby lowering computational costs and minimizing the energy demands associated with LLM inference. While we employ LLMs, we primarily use them for Question Answering on publicly available benchmarks and do not use any private information. We also would like to highlight that LLMs are known to hallucinate and may result in factually inaccurate answers at times, though we try to mitigate this in our approach by providing relevant context and by generating rationales.

## Contributions

Kiran Purohit and Venktesh V contributed equally to this work. Kiran played a major role in the conceptualization and implementation of the algorithm, conducted real-world experiments for CASE and baselines on Llama and Mistral, and played a major role in writing the paper. Venktesh V contributed to the algorithm's conceptualization and implementation, provided the sample complexity proof, conducted synthetic experiments, real-world baselines, GPT-4

experiments, and contributed to the writing of the paper. Sourangshu contributed to the formulation and writing of Section 3 and ideation of the proof. Sourangshu and Avishek mentored the project and contributed to the idea conceptualization, ideation, and writing of the paper.

## Acknowledgements

We thank the reviewers for their valuable and insightful feedback. Sourangshu Bhattacharya is thankful to ANRF Core Research Grant (File no: CRG/2023/004600), Govt. of India for supporting this project.

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

# A. Appendix

## A.1. Proof of lemma 1

*Proof.* We primarily follow the proof structure of GIFA framework (Réda et al., 2021) with some modifications required for CASE due to the shortlist $N_t$ and our swapping rule to compute $U_t$. Let $\mathcal{S}_m^\star$ be the true set of top-$m$ arms and $(S_m^*)^c$ denote the true set remaining worst arms. To prove Lemma 1, we introduce the following property,

**Property 1**: For $b_t \in U_t$ and $s_t \in N_t$ it holds that $\hat{\rho}_t(b_t) \geq \hat{\rho}_t(s_t)$. Hence, it follows that $B_t(s_t, b_t) = \hat{\Delta}_t(s_t, b_t) + W_t(b_t, s_t) \leq W_t(b_t, s_t)$ as $\hat{\Delta}_t(s_t, b_t) < 0$ From property 1, we can establish that $B_t(s_t, b_t) \leq W_t(b_t, s_t)$. Hence, to show that

$$B_t(s_t, b_t) \leq -(\Delta(b_t) \vee \Delta(s_t)) + 3W_t(b_t, s_t)$$

we consider the following scenarios:

**(i)** $b_t \in \mathcal{S}_m^\star$ **and** $s_t \notin \mathcal{S}_m^\star$: In that case,

$$\Delta(b_t) = \rho(b_t) - \rho(m+1); \Delta(s_t) = \rho(m) - \rho(s_t)$$

As event $\mathcal{E}$ holds,

$$\begin{aligned}
B_t(s_t, b_t) &= -B_t(b_t, s_t) + 2W_t(b_t, s_t) \\
&\leq \Delta(s_t, b_t) + 2W_t(b_t, s_t)
\end{aligned}$$

As $s_t \notin \mathcal{S}_m^\star$,
$$\rho(s_t) \leq \rho(m+1)$$
$$\Delta(s_t, b_t) \leq \rho(m+1) - \rho(b_t) = -\Delta(b_t)$$

But as $b_t \in \mathcal{S}_m^\star$, it also holds that $\rho(b_t) \geq \rho(m)$, and $\Delta(s_t, b_t) \leq \rho(s_t) - \rho(m) = -\Delta(s_t)$. Hence,

$$\begin{aligned}
B_t(s_t, b_t) &\leq -(\Delta(b_t) \vee \Delta(s_t)) + 2W_t(b_t, c_t) \\
&\leq -(\Delta(b_t) \vee \Delta(s_t)) + 3W_t(b_t, c_t).
\end{aligned}$$

**(ii)** $b_t \notin \mathcal{S}_m^\star$ **and** $s_t \in \mathcal{S}_m^\star$ :

$$\Delta(s_t) = \rho(s_t) - \rho(m+1); \Delta(b_t) = \rho(m) - \rho(b_t)$$

By Property 1,

$$\begin{aligned}
B_t(s_t, b_t) &\leq W_t(b_t, s_t) \\
&\leq \hat{\Delta}_t(b_t, s_t) + W_t(b_t, s_t) = B_t(b_t, s_t)
\end{aligned}$$

as $\hat{\rho}_t(b_t) \geq \hat{\rho}_t(s_t)$. Further, as $\mathcal{E}$ holds,

$$\begin{aligned}
B_t(b_t, s_t) &= -B_t(s_t, b_t) + 2W_t(b_t, s_t) \\
&\leq \Delta(b_t, s_t) + 2W_t(b_t, s_t)
\end{aligned}$$

As $b_t \notin \mathcal{S}_m^\star$, $\rho(b_t) \leq \rho(m+1)$ and hence $\Delta(b_t, s_t) \leq \rho(m+1) - \rho(s_t) = -\Delta(s_t)$ As $s_t \in \mathcal{S}_m^\star$, $\rho(s_t) \geq \rho(m)$ and hence $\Delta(b_t, s_t) \leq \rho(b_t) - \rho(m) = -\Delta(b_t)$. Hence,

$$\begin{aligned}
B_t(s_t, b_t) &\leq -(\Delta(b_t) \vee \Delta(s_t)) + 2W_t(b_t, c_t) \\
&\leq -(\Delta(b_t) \vee \Delta(s_t)) + 3W_t(b_t, c_t).
\end{aligned}$$

**(iii)** $b_t \notin \mathcal{S}_m^\star$ **and** $s_t \notin \mathcal{S}_m^\star$: We state that there exists a $b \in \mathcal{S}_m^\star$ that belongs to $N_t$. At any time t,

$$M_t \leftarrow s' \sim_{m'} (U_t \cup N_{t-1})^c$$

$$N_t \leftarrow \text{top}_{m'}(M_t \cup N_{t-1}; \hat{\rho}_{(t-1)})$$

Due to the above sampling approach adopted for $N_t$ which captures the next m' arms with the highest means, we posit that $N_t$ captures at least one arm in $\mathcal{S}_m^\star$. Assuming the event $\mathcal{E}$ holds and $b \in \mathcal{S}_m^\star$,

$$W_t(b_t, s_t) \geq B_t(s_t, b_t) \geq B_t(b, b_t)$$

$s_t$ by the definition is one of the most ambiguous arms with **largest gap to** $b_t$ $B_t(s_t, b_t) \geq B_t(b, b_t)$. Hence, $B_t(s_t, b_t) \geq B_t(b, b_t)$. From this and event $\mathcal{E}$ it follows

$$B_t(s_t, b_t) \geq B_t(b, b_t) \geq \rho(b) - \rho(b_t) \geq \rho(m) - \rho(b_t)$$

. Hence $W_t(b_t, s_t) \geq B_t(s_t, b_t) \geq \Delta(b_t)$. Using event $\mathcal{E}$,

$$\begin{aligned}
B_t(s_t, b_t) &\leq \Delta(s_t, b_t) + 2W_t(b_t, s_t) = (\rho(s_t) - \rho(m)) + \\
&\quad (\rho(m) - \rho(b_t)) + 2W_t(b_t, s_t)
\end{aligned}$$

From above Eq and since $B_t(s_t, b_t) \geq \Delta(b_t)$,

$$\begin{aligned}
B_t(s_t, b_t) &\leq -\Delta(s_t) + \Delta(b_t) + 2W_t(b_t, s_t) \\
&\leq -\Delta(s_t) + 3W_t(b_t, s_t)
\end{aligned}$$

Also from Property 1 and $W_t(b_t, s_t) \geq \Delta(b_t)$, it holds that

$$\begin{aligned}
B_t(s_t, b_t) &\leq W_t(b_t, s_t) = -W_t(b_t, s_t) + 2W_t(b_t, s_t) \\
&\leq -\Delta(b_t) + 2W_t(b_t, s_t) \leq -\Delta(b_t) + 3W_t(b_t, s_t)
\end{aligned}$$

Hence $B_t(s_t, b_t) \leq -(\Delta(b_t) \vee \Delta(s_t)) + 3W_t(b_t, c_t)$.

**(iv)** $b_t \in \mathcal{S}_m^\star$ **and** $s_t \in \mathcal{S}_m^\star$: Then there exists a $s \notin S_m^*$ and $s \in U_t$ In that case,

$$\Delta(b_t) = \rho(b_t) - \rho(m+1); \Delta(s_t) = \rho(s_t) - \rho(m+1)$$

Also by definition of $b_t$ and $s_t$, it holds that $B_t(s_t, b_t) = \max_{i \in U_t} \max_{j \in N_t} [B_t(j, i)]$ Since there exists $s \in U_t$ and $s_t \in N_t$,

$$B_t(s_t, b_t) = \max_{i \in U_t} \max_{j \in N_t} [B_t(j, i)] \geq \max_{j \in N_t} B_t(j, s)$$
$$\geq B_t(s_t, s) \geq \rho(s_t) - \rho(s) \geq \rho(s_t) - \rho(m+1)$$

As $\rho(s_t) - \rho(m+1) = \Delta(s_t)$, $B_t(s_t, b_t) \geq \Delta(s_t)$ By property 1, $B_t(s_t, b_t) \leq W_t(b_t, s_t)$. Hence,

$$\Delta(s_t) \leq B_t(s_t, b_t) \leq W_t(b_t, s_t)$$

On event $\mathcal{E}$ it follows that $B_t(s_t, b_t) \leq \rho(s_t) - \rho(b_t) + 2W_t(b_t, s_t)$ as $(B(s_t, b_t) \leq W_t(b_t, s_t)$. Then $\rho(s_t) - \rho(b_t)$ can be expressed as $\rho(s_t) - \rho(m+1) + \rho(m+1) - \rho(b_t)$. hence,

$$B_t(s_t, b_t) \leq \rho(s_t) - \rho(m+1) + \rho(m+1) - \rho(b_t)$$
$$+ 2W_t(b_t, s_t) \leq \Delta(s_t) - \Delta(b_t) + 2W_t(b_t, s_t)$$

We already know that $B_t(s_t, b_t) \geq \Delta(s_t)$ resulting in,

$$(a) \ B_t(s_t, b_t) \leq -\Delta(b_t) + 3W_t(b_t, s_t)$$

Now to prove $B_t(s_t, b_t) \leq -\Delta(s_t) + 3W_t(b_t, s_t)$, we rely on property 1,

$$B(s_t, b_t) \leq W_t(b_t, s_t) \leq -W_t(b_t, s_t) + 2W_t(b_t, s_t)$$

As $W_t(b_t, s_t) \geq \Delta(s_t)$, $-W_t(b_t, s_t) \leq -\Delta(s_t)$. Hence,

$$(b) \ B(s_t, b_t) \leq W_t(b_t, s_t) \leq -W_t(b_t, s_t) + 2W_t(b_t, s_t)$$
$$\leq -\Delta(s_t) + W_t(b_t, s_t) \leq -\Delta(s_t) + 3W_t(b_t, s_t)$$

From (a) and (b) $B_t(s_t, b_t) \leq -(\Delta(b_t) \vee \Delta(s_t)) + 3W_t(b_t, c_t)$. □

### A.2. Proof Structure for Theorem 1

*Proof.* Combining Lemma 4 with stopping rule $B_t(s_t, b_t) \leq \epsilon$ following Lemma 8 in (Réda et al., 2021) directly yields

$$N_{a_t}(t) \leq 4\sigma^2 C_{\delta,t}^2 \max\left(\varepsilon, \frac{\varepsilon + \Delta_{a_t}}{3}\right)^{-2}$$

where $N_{a_t}(t)$ is the number of times arms $a$ is sampled. This is equivalent to the sample complexity term $\mathrm{H}^\varepsilon(\mu)$ in Theorem 1. Hence, maximum number of samplings on event $\mathcal{E}$ is upper-bound by $\inf_{u \in \mathbb{R}^{*+}} \{u > 1 + \mathrm{H}^\varepsilon(\mu) C_{\delta,u}^2\}$, where $\mathrm{H}^\varepsilon(\mu) \triangleq 4\sigma^2 \sum_{a \in [K]} \max\left(\varepsilon, \frac{\varepsilon + \Delta_a}{3}\right)^{-2}$. □

### A.3. Datasets Description

An overview of the dataset statistics and examples are shown in Table 4.

**FinQA**: Comprises financial questions over financial reports that require numerical reasoning with structured and unstructured evidence. Here, 23.42% of the questions only require the information in the text to answer; 62.43% of the questions only require the information in the table to answer; and 14.15% need both the text and table to answer. Meanwhile, 46.30% of the examples have one sentence or one table row as the fact; 42.63% has two pieces of facts; and 11.07% has more than two pieces of facts. This dataset has 1147 questions in the evaluation set.

**AquaRat**: It comprises 100,000 algebraic word problems in the train set with dev and test set each comprising 254 problems. The problems are provided along with answers and rationales providing the step-by-step solution to the problem. An examples problem is shown in Table 4.

**TabMWP**: It is a tabular-based math word problem-solving dataset with 38,431 questions. TabMWP is rich in diversity, where 74.7% of the questions in TabMWP belong to free-text questions, while 25.3% are multi-choice. We evaluate on the test set with 7686 problems.

**GSM8K**: This dataset consists of linguistically diverse math problems that require multi-step reasoning. The dataset consists of 8.5K problems and we evaluate on the test set of 1319 questions.

**StrategyQA**: To prove the generality of our approach for reasoning tasks, we evaluate on StrategyQA (Geva et al., 2021), a dataset with implicit and commonsense reasoning questions. Since there is no public test set with ground truth answers, we perform stratified sampling done on 2290 full train set to split into 1800 train and 490 test.

**Metrics**: For TabMWP and StrategyQA we employ cover-EM (Rosset et al., 2021; Press et al., 2023), a relaxation of Exact Match metric which checks whether the ground truth answer is contained in the generated answer. This helps handle scenarios where LLM generates "24 kilograms" and the ground truth is "24". For other numerical reasoning datasets, we employ Exact match.

### A.4. Results using Alternate Open Source LLMs

We also report the performance of exemplars from CASE on open-source models like Mistral-7b and LLama2-7b. The results are shown in Table 6. We observe that the absolute performance across baselines and CASE is lower for smaller LLMs like Llama2 and Mistral-7b when compared to gpt-3.5-turbo or gpt-4o-mini. We observe that this is due to the scale of the Language models as Mistral and LlAMA2 models have 7 billion parameters while gpt-3.5-turbo is of

| Dataset | #Train | #Test | Example Question | Description |
|---|---|---|---|---|
| **GSM8K** (Cobbe et al., 2021) | 7473 | 1319 | `The red car is 40% cheaper than the blue car. The price of the blue car is $100. How much do both cars cost?` | multi-step arithmetic word problems |
| **AquaRat** (Ling et al., 2017) | 97467 | 254 | `John found that the average of 15 numbers is 40. If 10 is added to each number then the mean of number is?` | multi-step arithmetic word problems |
| **TabMWP** (Lu et al., 2023) | 23059 | 7686 | `A newspaper researched how many grocery stores there are in each town. What is the mean of the numbers?` | Table based numerical reasoning |
| **FinQA** (Chen et al., 2022b) | 6251 | 1147 | `what is the percentage change in the the gross liability for unrecognized tax benefits during 2008 compare to 2007?` | Table and Text based numerical reasoning |
| **StrategyQA** (Geva et al., 2021) | 1800 | 490 | `Does the United States Secretary of State answer the phones for the White House?` | multi-step reasoning |

Table 4: Overview of the Complex QA datasets used in this study.

much larger scale and the emergent capabilities like ICL, reasoning capabilities are more pronounced in large scale models (Wei et al., 2022).

However, we still observe that CASE leads to reasonable performance gains over other static exemplar selection methods across the smaller open-source LLMs. We also observe that CASE is competitive with instance-level/dynamic exemplar selection methods.

Our main experiments are carried out in an exemplar reuse setup where exemplar selection is done using small open source LLMs and transferred to larger LLMs. This is done to reduce the LLM inference cost during exemplar selection. This setup also leverages the reasoning and emergent capabilities of large scale LLMs. This philosophy is inspired from the work $\mu$P (Yang et al., 2022) where the language model hyperparameters are tuned using smaller LM and transferred to a larger LM for the task under consideration.

### A.5. Robustness of exemplars selected by CASE

We compare the robustness of CASE to other exemplar selection methods. We measure standard deviation of performance across different subsets of the evaluation set through 10-fold cross validation, as shown in Table 5. We observe that in 3 out of 4 datasets, exemplars chosen by CASE has

less variance in task performance when compared to other exemplar selection methods. Exemplars selected through instance-level approaches are not optimized for the task but rather on a per-test-example basis. Consequently, this leads to greater variance in final task performance. Hence, CASE helps select exemplars for the task which are more robust than other static methods or instance-level selection methods.

### A.6. Exemplar Reuse by CASE

For CASE, we evaluate the performance of gpt-3.5-turbo using exemplars selected by smaller models, such as Llama2-7b and Mistral-7b. Figure 4 shows the exemplars selected from Llama2-7b using CASE reused for gpt-3.5-turbo. In Table 1 we present the results of exemplar reuse from Mistral-7b for CASE. We observe that exemplars selected by CASE using smaller LLMs perform well with larger LLMs, promoting both exemplar reuse and efficiency.

### A.7. Prompts

We also demonstrate the instructions issued to the LLM for different tasks discussed in this work, along with some exemplars selected using CASE. An example of prompt

| Datasets | GSM8K | AquaRat | TabMWP | FinQA | StrategyQA |
|---|---|---|---|---|---|
| Zero-Shot COT | ±5.18 | ±7.08 | ±1.84 | ±4.50 | ±4.19 |
| Few-Shot COT | ±4.48 | ±12.03 | ±1.66 | ±4.76 | ±5.67 |
| KNN | ±3.76 | ±5.49 | ±1.27 | ±4.17 | ±4.85 |
| MMR | ±4.00 | ±10.53 | ±1.68 | ±6.10 | ±5.70 |
| Graph Cut | ±6.38 | ±8.18 | ±2.03 | ±5.29 | ±7.62 |
| Facility Location | ±4.23 | ±6.71 | ±1.74 | ±4.94 | ±5.93 |
| LENS | ±5.04 | ±6.67 | ±1.59 | ±5.81 | ±3.98 |
| **CASE** | ±**3.47** | ±6.86 | ±**0.88** | ±**3.72** | ±**2.91** |

Table 5: Comparison of robustness of CASE to other approaches. We report standard deviation (lower is better) with scores from different splits of the evaluation set.

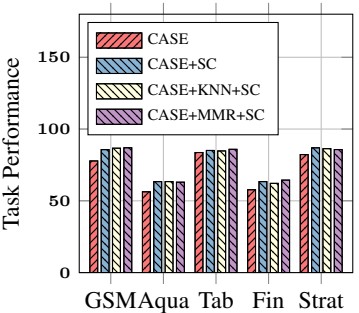

Figure 4: Reuse, Llama2 to gpt-3.5-turbo.

| Method | GSM8K | AquaRat | TabMWP | FinQA | StrategyQA |
|---|---|---|---|---|---|
| **Mistral-7B** | | | | | |
| **Instance-level** | | | | | |
| KNN (Rubin et al., 2022) | 28.00 | 23.16 | 45.3 | 9.06 | 78.27 |
| MMR (Ye et al., 2023b) | 28.97 | 18.11 | 47.61 | 10.11 | 79.95 |
| **Task-level** | | | | | |
| Zero-shot-COT (Kojima et al., 2023) | 7.42 | 18.89 | 38.96 | 1.74 | 35.37 |
| Manual Few-shot COT | 22.36 | 14.90 | 41.93 | 3.22 | 62.55 |
| LENS (Li & Qiu, 2023) | 26.08 | 14.17 | 41.82 | 5.14 | 76.12 |
| **Our Approach** | | | | | |
| CASE | **32.6** | 21.2 | **45.55** | **11.24** | 77.75 |
| **Llama2-7B** | | | | | |
| **Instance-level** | | | | | |
| KNN (Rubin et al., 2022) | 22.51 | 23.62 | 43.02 | 10.37 | 76.35 |
| MMR (Ye et al., 2023b) | 21.60 | 21.65 | 41.66 | 12.20 | 76.32 |
| **Task-level** | | | | | |
| Zero-shot-COT (Kojima et al., 2023) | 6.14 | 6.29 | 12.64 | 1.67 | 53.27 |
| Manual Few-shot COT | 19.26 | 20.47 | 23.62 | 2.87 | 64.29 |
| LENS (Li & Qiu, 2023) | 17.06 | 19.29 | 33.20 | 6.62 | 73.06 |
| **Our Approach** | | | | | |
| CASE | 21.91 | **24.02** | **44.69** | 9.59 | **77.55** |

Table 6: Results across datasets on MISTRAL-7B and LLAMA-2-7B (5-shot exemplars).

construction for FinQA is shown in Figure 9. We also showcase example prompts for AquaRat (Figure 8), GSM8K (Figure 7), TabMWP (Figure 10) and StrategyQA (Figure 11).

### A.8. Exemplar Qualitative Analysis

We provide a qualitative analysis of exemplars and compare the exemplars selected using CASE with exemplars selected using LENS (Li & Qiu, 2023), the recent state-of-the-art approach. The final set of exemplars chosen by LENS vs CASE for the AquaRat dataset is shown in Table 7. We observe that Question 4 and Question 5 in the set of exemplars chosen by LENS are redundant in that they are very similar problems that require similar reasoning steps and are also similar thematically. Both the questions are centered on the theme of work and time and are phrased in a similar manner. Hence, they do not add any additional information to solve diverse problems the LLM may encounter during inference. However, we observe that the exemplars chosen by CASE are problems that require diverse reasoning capabilities and are also different thematically.

We also compare the exemplars chosen by CASE with LENS for the FinQA dataset. We observe that the exemplars chosen by CASE comprises diverse set of problems. We also observe that CASE also contains exemplars that require

composite numerical operations with multi-step reasoning rationales to arrive at the solutions, whereas LENS mostly has exemplars with single-step solutions.

The exemplars chosen by LENS compared to CASE for TabMWP are shown in Table 10. We observe that exemplar 1 and exemplar 3 chosen by LENS are redundant, as they represent the same reasoning concept of computing median for a list of numbers. However, we observe that CASE selects diverse exemplars, with each exemplar representing a different reasoning concept. We also demonstrate the exemplars for GSM8K and StrategyQA in Table 9 and Table 11 respectively.

### A.9. Synthetic Experiments - Efficiency and Convergence Analysis

We further present the results for synthetic experiments for scenarios where $K = 7, m = 3, N = 3$ and $K = 10, m = 3, N = 3$ in Figures 5 and 6. We observe that in both the cases, CASE **drastically reduces** the number of gap-index computations and comparison based operations (comparing arms). For instance, for $K = 10, m = 3$ scenario, on average across 500 simulations CASE only requires 20366.84 comparisons, whereas LinGapE requires 948206.10 comparisons and LinGIFA requires 2180251.73 comparisons. This is due to the shortlist of challenger arms $N_t$ maintained

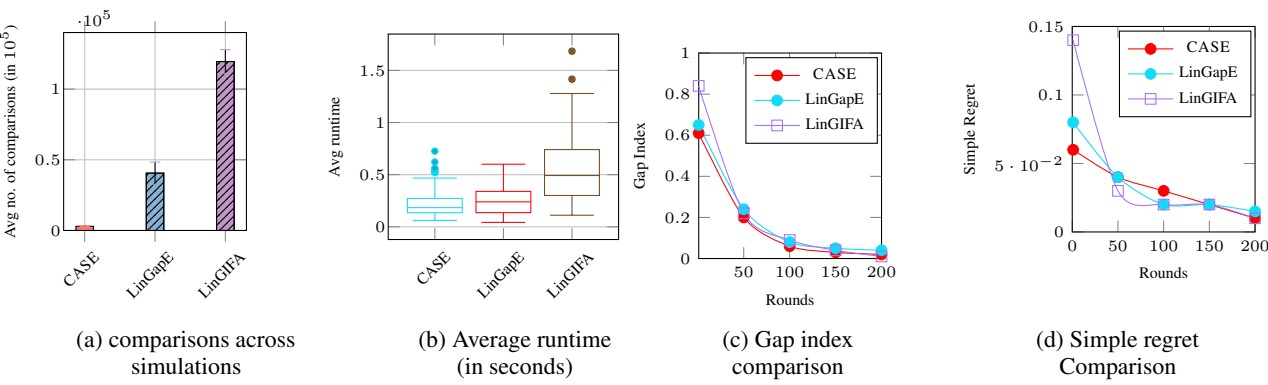

Figure 5: Top-$m$ arm identification by CASE, LinGIFA and LinGapE for K=7, m=3, N=3. (a) Average number of comparisons across simulations (b) Average runtime (in seconds) (c) Gap Index ($B_t(s_t, b_t)$) comparison and (d) Simple regret comparison for each round across simulations

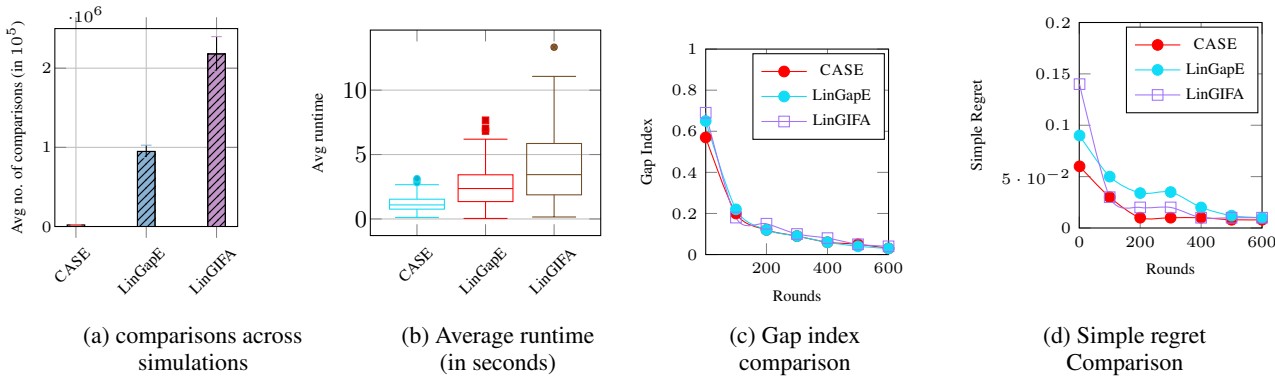

Figure 6: Top-$m$ arm identification by CASE, LinGIFA and LinGapE for K=10, m=3, N=3. (a) Average number of comparisons across simulations (b) Average runtime (in seconds) (c) Gap Index ($B_t(s_t, b_t)$) comparison and (d) Simple regret comparison for each round across simulations

by the proposed approach CASE. We also observe that this results in significant reduction in time (approx. **6x** lower compared to LinGIFA and **2.5x** compared to LinGapE for $K = 10$ case) due to low number of comparisons. We observe that the gap index and simple regret approaches 0 in a similar trend for all algorithms. This demonstrates that CASE converges with much lower gap index computations and has lower runtime compared to existing state-of-the-art gap index algorithms.

---

**GSM8K Prompt**

**Instruction:** You are a helpful, respectful and honest assistant helping solve math word tasks requiring reasoning. Follow the given examples and solve the tasks in step by step manner.

**Exemplars** :
[Question]: *The red car is 40% cheaper than the blue car. The price of the blue car is $100. How much do both cars cost?*
[Explanation]: The red car is $40/100 * 100 = 40$ cheaper than the blue car.
That means, that the red car costs $100 - 40 = 60$.
So both cars cost $100 + 60 = 160$
[Answer]: 160
...

**Test Input** : Question: Explanation: [INS] Answer: [INS]

Figure 7: Prompt for GSM8K

---

**AQUA Prompt**

**Instruction:** You are a helpful, respectful and honest assistant helping solve math word tasks requiring reasoning. Follow given examples and solve the tasks in step by step manner.

**Exemplars** :
[Question]: *John found that the average of 15 numbers is 40. If 10 is added to each number then the mean of the number is?*
[Options]: A) 50, B) 45, C) 65, D) 78, E) 64
[Explanation]: $(x0 + x1 + ...x14)/15 = 40$,
new_mean = $40 + 10 = 50$
[Answer]: The option is A
...

**Test Input** : Question: Options: Explanation: [INS] Answer: [INS]

Figure 8: Prompt for AquaRat

---

**FinQA Prompt**

**Instruction:** You are a helpful, respectful and honest assistant helping solve math word tasks requiring reasoning, using the information from given table and text.

**Exemplars** :
*Read the following table, and then answer the question:*
[Table]: beginning balance as of december 1 2007 | 201808
gross increases in unrecognized tax benefits 2013 prior year tax positions | 14009
gross increases in unrecognized tax benefits 2013 current year tax positions | 11350
ending balance as of november 28 2008 | 139549
[Question]: *what is the percentage change in the the gross liability for unrecognized tax benefits during 2008 compare to 2007?*
[Explanation]: x0 = $139549 - 201808$,
ans = $x0/201808$
[Answer]: -30.9%
...

**Test Input:** Read the table and answer the question: Table: Question: Explanation: [INS] Answer: [INS]

Figure 9: Prompt for FinQA

---

**TabMWP Prompt**

**Instruction**:You are a helpful, respectful and honest assistant helping to solve math
word problems or tasks requiring reasoning or math, using the information from the
given table.  Solve the given problem step by step providing an explanation for your
answer.

**Exemplars**  :
[Table]: Town | Number of stores
Mayfield | 9
Springfield | 9
Riverside | 6
Chesterton | 5
Watertown | 2
[Question]: *A newspaper researched how many grocery stores there are in each town. What is the range of the numbers?*
[Explanation]: Read the numbers from the table. $9, 9, 6, 5, 2$
First, find the greatest number. The greatest number is $9$.
Next, find the least number. The least number is $2$.
Subtract the least number from the greatest number: $9 - 2 = 7$
[Answer]: The range is 7
. . .
. . .

**Test Input**   : Table:  Question:
Explanation: [INS] Answer: [INS]

---

Figure 10: Prompt for TabMWP

---

**StrategyQA Prompt**

**Instruction**:You are a helpful, respectful and honest assistant helping to solve
commonsense problems requiring reasoning.  Follow the given examples that use
the facts to answer a question by decomposing into sub-questions first and then
predicting the final answer as "Yes" or "No" only.

**Exemplars**  :
[Facts]: The role of United States Secretary of State carries out the President's foreign policy. The White House has multiple
phone lines managed by multiple people.
[Question]: *Does the United States Secretary of State answer the phones for the White House?*
[Sub-question 1]: What are the duties of the US Secretary of State?
[Sub-question 2]: Are answering phones part of #1?

[Answer]: No
. . .
. . .

**Test Input**   : Facts:  Question:
Sub-question: [INS] Answer: [INS]

---

Figure 11: Prompt for StrategyQA

| Method | Exemplars |
|---|---|
| LENS | **Question**: A cat chases a rat 6 hours after the rat runs. cat takes 4 hours to reach the rat. If the average speed of the cat is 90 kmph, what s the average speed of the rat? 
 **Options**: ['A)32kmph', 'B)26kmph', 'C)35kmph', 'D)36kmph', 'E)32kmph'] 
 **Rationale:** Cat take 10 hours and rat take 4 hours...then Distance is 90*4.so speed of rat is (90*4)/10 = 36kmph **Answer: D** |
| | **Question:** A business executive and his client are charging their dinner tab on the executive's expense account.The . . . ? 
 **Options**: ['A)69.55\$', 'B)50.63\$', 'C)60.95\$', 'D)52.15\$', 'E)53.15'] 
 **Rationale**: let x is the cost of the food 1.07x is the gross bill after including sales tax 1.15* 1.07x=75 **Answer**: C |
| | **Question:**John and David were each given X dollars in advance for each day they were expected to perform at a community festival. John eventually,. . . ? 
 **Options**: 'A)11Y', 'B)15Y', 'C)13Y', 'D)10Y', 'E)5Y' **Rationale:** . . . **Answer:** A |
| | **Question:**A contractor undertakes to do a piece of work in 40 days. He engages 100 men at the beginning and 100 more after 35 days and completes the work in stipulated time. If he had not engaged the additional men, how many days behind schedule would it be finished?? 
 **Options**: 'A)2', 'B)5', 'C)6', 'D)8', 'E)9' **Rationale:** [(100× 35)+(200× 5)] men can finish the work in 1 day therefore 4500 men can finish the work in 1 day. 100 men can finish it in $\frac{4500}{100}$ = 45 days. This is 5 days behind Schedule **Answer:** A |
| | **Question:** A can do a job in 9 days and B can do it in 27 days. A and B working together will finish twice the amount of work in ——- days? 
 **Options**: 'A)22 days', 'B)18 days', 'C)22 6/2 days', 'D)27 days', 'E)9 days' **Rationale:** 1/9 + 1/27= 3/27 = 1/9 9/1 = 9*2 =18 day **Answer:** B |
| CASE | **Question**: In a 1000 m race, A beats B by 50 m and B beats C by 100 m. In the same race, by how many meters does A beat C? 
 **Options**: 'A)156 m', 'B)140 m', 'C)145 m', 'D)169 m', 'E)172 m' **Rationale:** By the time A covers 1000 m, B covers (1000 - 50) = 950 m. By the time B covers 1000 m, C covers (1000 - 100) = 900 m. So, the ratio of speeds of A and C = 1000/950 * 1000/900 = 1000/855. So, by the time A covers 1000 m, C covers 855 m. So in 1000 m race A beats C by 1000 - 855 = 145 m. **Answer: C** 
 **Question:** Count the numbers between 10 - 99 which yield a remainder of 3 when divided by 9 and also yield a remainder of 2 when divided by 5? **Options**: 'A)Two', 'B)Five', 'C)Six', 'D)Four', 'E)One' 
 **Rationale**: Numbers between 10 - 99 giving remainder 3 when divided by 9 = 12, 21, 30, 39, 48, 57, 66, 75, 84, 93. The Numbers giving remainder 2 when divided by 5 = 12, 57 = 2 **Answer**: A 
 **Question:** A train running at the speed of 60 km/hr crosses a pole in 3 seconds. Find the length of the train. **Options**: 'A)60', 'B)50', 'C)75', 'D)100', 'E)120' **Rationale:** Speed = 60*(5/18) m/sec = 50/3 m/sec. Length of Train (Distance) = Speed * Time (50/3) * 3 = 50 meter. **Answer:** B 
 **Question:** If n is an integer greater than 7, which of the following must be divisible by 3? 
 **Options**: 'A)1. n (n+1) (n-4)', 'B)2. n (n+2) (n-1)', 'C)3. n (n+3) (n-5)', 'D)4. n (n+4) (n-2)', 'E)5. n (n+5) (n-6)' **Rationale:** We need to find out the number which is divisible by three, In every 3 consecutive integers, there must contain 1 multiple of 3. So n+4 and n+1 are same if we need to find out the 3's multiple. replace all the numbers which are more than or equal to three . . . **Answer:** D 
 **Question:** A merchant gains or loses, in a bargain, a certain sum. In a second bargain, he gains 280 dollars, and, in a third, loses 20. In the end he finds he has gained 120 dollars, by the three together. How much did he gain or lose bv the first ? **Options**: 'A)80', 'B)-140', 'C)140', 'D)120', 'E)None' 
 **Rationale:** In this sum, as the profit and loss are opposite in their nature, they must be distinguished by contrary signs. If the profit is marked +, the loss must be -. Let x = the sum required. Then according to the statement x + 280 - 20 = 120. And x = -140. **Answer:** B |

Table 7: Qualitative analysis of exemplars for **AquaRat** dataset selected by LENS vs CASE. Rationale is not completely shown for some questions to conserve space. However, in our experiments all exemplars include rationales.

| Method | Exemplars |
|---|---|
| LENS | **Table**: \| increase ( decrease ) \| average yield \| 2.75% ( 2.75 % ) \| volume \| 0.0 to 0.25 \| energy services \| 2013 \| fuel recovery fees \| 0.25 \| recycling processing and commodity sales \| 0.25 to 0.5 \| acquisitions / divestitures net \| 1.0 \| total change \| 4.25 to 4.75% ( 4.75 % ) \|
**Question**: what is the ratio of the acquisitions / divestitures net to the fuel recovery fees as part of the expected 2019 revenue to increase
**Rationale**: ans=( 1.0 / 0.25 ) **Answer:** The answer is 4 |
| | **Table:** ( in millions ) \| 2009 \| 2008 \| 2007 \| sales and transfers of oil and gas produced net of production andadministrative costs \| -4876 ( 4876 ) \| -6863 ( 6863 ) \| -4613 ( 4613 ) \| . . .
**Question**: were total revisions of estimates greater than accretion of discounts?
**Rationale**: . . . **Answer**: The answer is yes |
| | **Table:** \| 2007 \| 2008 \| change \| capital gain distributions received \| 22.1 \| 5.6 \| -16.5 ( 16.5 ) \| other than temporary impairments recognized \| -.3 ( .3 ) \| -91.3 ( 91.3 ) \| -91.0 ( 91.0 ) \| net gains ( losses ) realized onfund dispositions \| 5.5 \| -4.5 ( 4.5 ) \| -10.0 ( 10.0 ) \| net gain ( loss ) . . .
**Question**: what percentage of tangible book value is made up of cash and cash equivalents and mutual fund investment holdings at december 31 , 2009? **Rationale**: ( 1.4 / 2.2 ) **Answer:** The answer is 64% |
| | **Table:** in millions \| 2009 \| 2008 \| 2007 \| sales \| 5680 \| 6810 \| 6530 \| operating profit \| 1091 \| 474 \| 839 \|
**Question**: north american printing papers net sales where what percent of total printing paper sales in 2009?
**Rationale**: x0=( 2.8 * 1000 ), ans=( x0 * 5680 ) **Answer:** The answer is 49% |
| | **Table:** in millions \| december 312015 \| december 312014 \| total consumer lending \| 1917 \| 2041 \| total commercial lending \| 434 \| 542 \| total tdrs \| 2351 \| 2583 \| nonperforming \| 1119 \| 1370 \| . . .
**Question**: what was the change in specific reserves in alll between december 31 , 2015 and december 31 , 2014 in billions? **Rationale**: ( .3 - .4 ) **Answer:** The answer is -0.1 |
| CASE | **Table**: in millions \| total \| balance december 31 2006 \| $ 124 \| payments \| -78 ( 78 ) \| balance december 31 2007 \| 46 \| additional provision \| 82 \| payments \| -87 ( 87 ) \| balance december 31 2008 \| 41 \| payments \| -38 ( 38 ) \| balance december 31 2009 \| $ 3 \|
**Question**: in 2006 what was the ratio of the class a shares and promissory notes international paper contributed in the acquisition of borrower entities interest **Rationale**: ans=( 200 / 400 ) **Answer:** 0.5 |
| | **Table:** \| 2018 \| 2017 \| 2016 \| allowance for other funds used during construction \| $ 24 \| $ 19 \| $ 15 \| allowance for borrowed funds used during construction \| 13 \| 8 \| 6 \|
**Question**: by how much did allowance for other funds used during construction increase from 2016 to 2018?
**Rationale**: x0=( 24 - 15 ),ans=( x0 / 15 ) **Answer**: 60% |
| | **Table:** ( dollars in millions ) \| 2001 ( 1 ) \| 2000 \| 1999 ( 2 ) \| change 00-01 \| adjusted change 00-01 ( 3 ) \| servicing fees \| $ 1624 \| $ 1425 \| $ 1170 \| 14% ( 14 % ) \| 14% ( 14 % ) \| management fees \| 511 \| 581 \| 600 \| -12 ( 12 ) \| -5 ( 5 ) \| foreign exchange trading \| 368 \| 387 \| 306 \| -5 ( 5 ) \| -5 ( 5 ) \| processing fees and other \| 329 \| 272 \| 236 \| 21 \| 21 \| total fee revenue \| 2832\| 2665 \| $ 2312 \| 6 \| 8 \| **Question**: what is the growth rate in total fee revenue in 2001? **Rationale**: x0=( 2832 - 2665 ),ans=( x0 / 2665 ) **Answer:** 6.30% |
| | **Table:** \| increase (decrease) \| average yield \| 2.75% (2.75 %) \| volume \| 0.0 to 0.25 \| energy services \| 2013 \| fuel recovery fees \| 0.25 \| recycling processing and commodity sales \| 0.25 to 0.5 \| acquisitions / divestitures net \| 1.0 \| total change \| 4.25 to 4.75% ( 4.75 % ) \| **Question**: what is ratio of insurance recovery to incremental cost related to our closed bridgeton landfill **Rationale**: ans=(40.0/12.0) **Answer:** 3.33 |
| | **Table:** $ in millions \| as of december 2018 \| as of december 2017 \| fair value of retained interests \| $ 3151 \| $ 2071 \| weighted average life ( years ) \| 7.2 \| 6.0 \| constant prepayment rate \| 11.9% ( 11.9 % ) \| 9.4% ( 9.4 % ) \| impact of 10% ( 10 % ) adverse change \| $ -27 ( 27 ) \| $ -19 ( 19 ) \| impact of 20% ( 20 % ) adverse change \| $ -53 ( 53 ) \| $ -35 ( 35 ) \| discount rate \| 4.7% ( 4.7 % ) \| 4.2% ( 4.2 % ) \| impact of 10% ( 10 % ) adverse change \| $ -75 ( 75 ) \| $ -35 ( 35 ) \| impact of 20% ( 20 % ) adverse change \| $ -147 ( 147 ) \| $ -70 ( 70 ) \| **Question**: what was the change in fair value of retained interests in billions as of december 2018 and december 2017? **Rationale**: ans=( 3.28 - 2.13 ) **Answer:** 1.15 |

Table 8: Qualitative analysis of exemplars for **FinQA** dataset selected by LENS vs CASE. Rationale is not completely shown for some questions to conserve space. However, in our experiments all exemplars include rationales.

| Method | Exemplars |
|--------|-----------|
| LENS | **Question**: Michael wants to dig a hole 400 feet less deep than twice the depth of the hole that his father dug. The father dug a hole at a rate of 4 feet per hour. If the father took 400 hours to dig his hole …? 
 **Rationale:** Since the father dug a hole with a rate of 4 feet per hour, if the father took 400 hours digging the hole, he dug a hole 4*400 = 1600 feet deep. …Michael will have to work for 2800/4 = 700 hours. **Answer:** 700 |
| | **Question:** When Erick went to the market to sell his fruits, he realized that the price of lemons had risen by $4$ for each lemon. The price of grapes had also increased by half the price that …? 
 **Rationale**: The new price for each lemon after increasing by $4$ is $8 + 4 = 12$ For the 80 lemons, …Erick collected $140 * 9 = 1260$ From the sale of all of his fruits, Erick received $1260 + 960 = 2220$. **Answer**: 2220 |
| | **Question:**James decides to build a tin house by collecting 500 tins in a week. On the first day, he collects 50 tins. On the second day, he manages to collect 3 times that number. …? 
 **Rationale:** On the second day, he collected 3 times the number of tins he collected on the first day, which is $3 * 50 = 150$ tins. …he'll need to collect $200/4 = 50$ tins per day to reach his goal. 
 **Answer:** 50 |
| | **Question:** Darrel is an experienced tracker. He can tell about an animal by the footprints it leaves behind. Based on the impressions, he could tell the animal was traveling east at 15 miles/hour …? 
 **Rationale:** If we let x be the amount of time, in hours, it will take for Darrel to catch up to the coyote, …If we subtract 1 x from each side, we get x=1, the amount of time in hours. **Answer:** 1 |
| | **Question:** Martha needs to paint all four walls in her 12 foot by 16 foot kitchen, which has 10 foot high ceilings …If Martha can paint 40 square feet per hour, how many hours will it take her to paint kitchen? 
 **Rationale:** There are two walls that are 12' by 10' and two walls that are 16' by 10' …how many hours she needs to finish: 1680 sq ft / 40 sq ft/hour = 42 hours **Answer:** 42 |
| CASE | **Question**: Each class uses 200 sheets of paper per day. The school uses a total of 9000 sheets of paper every week. If there are 5 days of school days, how many classes are there in the school? 
 **Rationale:** Each class uses 200 x 5 = 1000 sheets of paper in a week. Thus, there are 9000/1000 = 9 classes in the school. **Answer:** 9 
 **Question:** If Jenna has twice as much money in her bank account as Phil does, and Phil has one-third the amount of money that Bob has in his account, and Bob has $60 in his account, how much less money does Jenna have in her account than Bob? **Rationale**: If Phil has one-third of the amount that Bob does, so he has $60/3= $20 in his account. Since Jenna has twice as much money as Phil, so she has $20*2= 40 in her account. Since Bob has $60 in his account, so he has $60-$40=$20 more than Jenna. **Answer**: 20 
 **Question:**Carlos bought a box of 50 chocolates. 3 of them were caramels and twice as many were nougats. The number of truffles was equal to the number of caramels plus 6. …If Carlos picks a chocolate at random, what is the percentage chance it will be a peanut cluster? **Rationale:** First find the number of nougats by doubling the number of caramels: 3 caramels * 2 nougats/caramel = 6 nougats. Then find the number of truffles by adding 7 to the number of caramels: 3 caramels + 6 = 9 …**Answer:** 64 
 **Question:** Janet has 60 less than four times as many siblings as Masud. Carlos has 3/4 times as many siblings as Masud. If Masud has 60 siblings, how many more siblings does Janet have more than Carlos? **Rationale:** If Masud has 60 siblings, and Carlos has 3/4 times as many siblings as Masud, Carlos has 3/4*60=45 siblings. Four times as many siblings as Masud has is 4*60=240. Janet has 60 less than four times as many siblings as Masud, a total of 240-60=180 siblings. …180-45=135 **Answer:** 135 
 **Question:** Gavin has had 4 dreams every day for a year now. If he had twice as many dreams last year as he had this year, calculate the total number of dreams he's had in the two years. **Rationale:** If Gavin has been having 4 dreams every day for a year now, he has had 4*365 = 1460 dreams this year. Gavin had twice as many dreams last as he had this year, meaning he had 2*1460 = 2920 dreams last year. The total number of dreams he has had in the two years is 2920+1460=4380 dreams. **Answer:** 4380 |

Table 9: Qualitative analysis of exemplars for **GSM8K** dataset selected by LENS vs CASE. Rationale is not completely shown for some questions to conserve space. However, in our experiments all exemplars include rationales.

| Method | Exemplars |
|---|---|
| LENS | **Table**: \| Name \| Age (years) \| Jessica \| 2 \| Dalton \| 7 \| Kelsey \| 5 \| Lamar \| 8 \| Alexis \| 2 **Question**: A girl compared the ages of her cousins. What is the median of the numbers? **Rationale:** Read the numbers from the table: 2, 7, 5, 8, 2. First, arrange the numbers from least to greatest: 2, 2, 5, 7, 8. Now find the number in the middle. The number in the middle is 5. The median is 5. **Answer:** 5 |
| | **Table:** \| City \| Number of houses sold \| Melville \| 878 \| New Hamburg \| 871 \| Charles Falls \| 881 \| Pennytown \| 817 **Question**: A real estate agent looked into how many houses were sold in different cities. Where were the fewest houses sold? **Rationale**: Find the least number in table. …The least number is 817. Now find the corresponding city. Pennytown corresponds to 817. **Answer**: 817 |
| | **Table:** \| Day \| Number of new customers \| Saturday \| 2 \| Sunday \| 2 \| Monday \| 9 \| Tuesday \| 4 \| Wednesday \| 10 \| Thursday \| 3 \| Friday \| 6 **Question**: A cable company analyst paid attention to how many new customers it had each day. What is the median of the numbers? **Rationale:** …Find the number in the middle. The number in the middle is 4. The median is 4. **Answer:** 4 |
| | **Table:** \| Day \| Number of cups \| Friday \| 8 \| Saturday \| 4 \| Sunday \| 10 \| Monday \| 6 \| Tuesday \| 6 \| Wednesday \| 1 \| Thursday \| 0 **Question**: Nancy wrote down how many cups of lemonade she sold in the past 7 days. What is the range of the numbers? **Rationale:** Read the numbers from the table: 8, 4, 10, 6, 6, 1, 0. …Subtract the least number from the greatest number: 10-0=10. The range is 10. **Answer:** 10 |
| | **Table:** \| Price \| Quantity demanded \| Quantity supplied \|$700 \| 9,800 \| 22,600 \|$740 \| 8,000 \| 22,800 \|$780 \| 6,200 \| 23,000 \|$820 \| 4,400 \| 23,200 \|$860 \| 2,600 \| 23,400 **Question**: At a price of $860, is there a shortage or a surplus? **Rationale:** At price of $860, quantity demanded is less than quantity supplied. …So, there is a surplus. **Answer:** surplus |
| CASE | **Table**: Number of siblings \| Frequency 0 \| 19 1 \| 12 2 \| 13 3 \| 9 **Question**: The students in Mr. Robertson's class recorded the no. of siblings that each has. How many students have fewer than 2 siblings? **Rationale:** Find the rows for 0 and 1 sibling. Add the frequencies for these rows. 19 + 12 = 31, 31 students have fewer than 2 siblings. **Answer:** 31 |
| | **Table:** \| Apples \| Peaches Organic \| 2 \| 7 Non-organic \| 7 \| 3 **Question**: Brittany conducted a blind taste test on some of her friends in order to determine if organic fruits tasted different than non-organic fruits. Each friend ate one type of fruit. What is the probability that a randomly selected friend preferred organic and tasted peaches? **Rationale**: Let A be the event "the friend preferred organic" and B be the event "the friend tasted peaches" …**Answer**: Jul-19 |
| | **Table:** dance performance ticket \| $29.00 play ticket \| $32.00 figure skating ticket \| $41.00 ballet ticket \| $37.00 opera ticket \| $76.00 orchestra ticket \| $58.00 **Question**: How much money does Hannah need to buy a ballet ticket and 7 orchestra tickets? **Rationale:** Find the cost of 7 orchestra tickets. $58.00 * 7 = $406.00 . Now find the total cost. $37.00 + $406.00 = $443.00. Hannah needs $443.00. **Answer:** 443 |
| | **Table:** Price \| Quantity demanded \| Quantity supplied $665 \| 15,500 \| 16,200 $855 \| 13,700 \| 17,300 $1,045 \| 11,900 \| 18,400 $1,235 \| 10,100 \| 19,500 $1,425 \| 8,300 \| 20,600 **Question**: Look at the table. Then answer the question. At a price of $1,045, is there a shortage or a surplus? **Rationale:** At the price of $1,045, the quantity demanded is less than the quantity supplied. There is too much of the good or service for sale at that price. So, there is a surplus. **Answer:** surplus |
| | **Table:** Comfy Pillows Resort \| 4:15 A.M \| 2:30 P.M \| 10:00 P.M Skyscraper City \| 4:45 A.M \| 3:00 P.M \| 10:30 P.M Pleasant River Campground \| 5:15 A.M \| 3:30 P.M \| 11:00 P.M Rollercoaster Land \| 5:45 A.M \| 4:00 P.M \| 11:30 P.M Floral Gardens \| 6:45 A.M \| 5:00 P.M \| 12:30 A.M Chickenville \| 7:15 A.M \| 5:30 P.M \| 1:00 A.M Happy Cow Farm \| 7:45 A.M \| 6:00 P.M \| 1:30 A.M **Question**: Look at the following schedule. Marshall got on the train at Rollercoaster Land at 5.45 A.M. What time will he get to Floral Gardens? **Rationale:** Find 5:45 A.M. in the row for Rollercoaster Land. That column shows the schedule for the train that Marshall is on. Look down the column until you find the row for Floral Gardens. Marshall will get to Floral Gardens at 6:45 A.M. **Answer:** 6:45 A.M. |

Table 10: Qualitative analysis of exemplars for **TabMWP** dataset selected by LENS vs CASE. Rationale is not completely shown for some questions to conserve space. However, in our experiments all exemplars include rationales.

| Method | Exemplars |
|---|---|
| LENS | **Facts**: Penguins are native to the deep, very cold parts of the southern hemisphere. Miami is located in the northern hemisphere and has a very warm climate.
**Question:** Would it be common to find a penguin in Miami?
**Rationale:** Where is a typical penguin's natural habitat? What conditions make #1 suitable for penguins? Are all of #2 present in Miami? **Answer: No** |
| | **Facts:** Shirley Bassey recorded the song Diamonds are Forever in 1971. Over time, diamonds degrade and turn into graphite. Graphite is the same chemical composition found in pencils.
**Question:** Is the title of Shirley Bassey's 1971 diamond song a true statement? **Rationale**: What is the title to Shirley Bassey's 1971 diamond song? Do diamonds last for the period in #1? **Answer**: No |
| | **Facts:** The first six numbers in the Fibonacci sequence are 1,1,2,3,5,8. Since 1 is doubled, there are only five different single digit numbers. **Question:** Are there five different single-digit Fibonacci numbers?
**Rationale:** What are the single-digit numbers in the Fibonacci sequence? How many unique numbers are in #1? Does #2 equal 5? **Answer:** Yes |
| | **Facts:** Katy Perry's gospel album sold about 200 copies. Katy Perry's most recent pop albums sold over 800,000 copies. **Question:** Do most fans follow Katy Perry for gospel music? **Rationale:** What type of music is Katy Perry known for? Is Gospel music the same as #1? **Answer:** No |
| | **Facts:** The Italian Renaissance was a period of history from the 13th century to 1600. A theocracy is a type of rule in which religious leaders have power. Friar Girolamo Savonarola was the ruler of Florence, after driving out the Medici family, from November 1494 â€ 23 May 1498. **Question:** Was Florence a Theocracy during Italian Renaissance? **Rationale:** When was the Italian Renaissance?When did Friar Girolamo Savonarola rule Florence? Is #2 within the span of #1? Did Friar Girolamo Savonarola belong to a religious order during #3? **Answer:** Yes |
| CASE | **Facts**: U2 is an Irish rock band that formed in 1976. The Polo Grounds was a sports stadium that was demolished in 1964. **Question**: Did U2 play a concert at the Polo Grounds? **Rationale:** When was U2 (Irish rock band) formed? When was the Polo Grounds demolished? Is #1 before #2? **Answer:** No
**Facts**: The capacity of Tropicana Field is 36,973. The population of Auburn, NY is 27,687. **Question:** Can you fit every resident of Auburn, New York, in Tropicana Field? **Rationale**: What is the capacity of Tropicana Field? What is the population of Auburn, NY? Is #1 greater than #2? **Answer**: Yes |
| | **Facts**: Door to door advertising involves someone going to several homes in a residential area to make sales and leave informational packets. . . . **Question:** During the pandemic, is door to door advertising considered inconsiderate? **Rationale:** What does door to door advertising involve a person to do? During the COVID-19 pandemic, what does the CDC advise people to do in terms of traveling? . . . Does doing #1 go against #2 and #3? **Answer:** Yes |
| | **Facts:** Mosquitoes cannot survive in the climate of Antarctica. Zika virus is primarily spread through mosquito bites. **Question:** Do you need to worry about Zika virus in Antarctica? **Rationale:** What animal spreads the Zika Virus? What is the climate of Antarctica? Can #1 survive in #2? **Answer:** No |
| | **Facts:** Bob Marley had 9 children. Kublai Khan had 23 children. Many of Bob Marley's children became singers, and followed his themes of peace and love. The children of Kublai Khan followed in his footsteps and were fierce warlords. **Question:** Could Bob Marley's children hypothetically win tug of war against Kublai Khan's children? **Rationale:** How many children did Bob Marley have? How many children did Kublai Khan have? Is #1 greater than #2? **Answer:** No |

Table 11: Qualitative analysis of exemplars for **StrategyQA** dataset selected by LENS vs CASE. Rationale is not completely shown for some questions to conserve space. However, in our experiments all exemplars include rationales.

