# OpenReview forum: "Sample Efficient Demonstration Selection for In-Context Learning"
_ICML.cc/2025/Conference — ICML 2025 poster_

### Official Review · Reviewer_xSam · 2025-03-10

**Overall Recommendation:** 4

**Summary:**

The paper presents a novel and efficient method for selecting demonstration examples in In-Context Learning (ICL). The proposed method, CASE, is shown to outperform existing exemplar selection techniques by significantly reducing the number of LLM calls, improving efficiency, and maintaining task performance.

**Claims And Evidence:**

Yes.

**Essential References Not Discussed:**

No.

**Experimental Designs Or Analyses:**

Yes.

**Methods And Evaluation Criteria:**

Yes.

**Other Comments Or Suggestions:**

Please refer to the Strengths and Weaknesses.

**Other Strengths And Weaknesses:**

Strength:

Totally, the proposed approach seems theoretically sound and well-grounded in multi-armed bandit (MAB) frameworks, which is effectively applied to the exemplar selection problem. More important, I support the current design, which can effectively reduce the calls to LLM, and this issue has practical significance. It can reduce the data selection cost by reducing the number of API calls. Experimental results also demonstrate impressive efficiency gains, particularly in terms of reducing LLM calls and improving runtime, compared to state-of-the-art methods like LENS and EXPLORA.

Weakness and Question:

1. How could the current method be applied to machine translation in ICL?[1] Specifically, how can it be used to select suitable samples to improve translation accuracy?

[1] In-context Examples Selection for Machine Translation.

2. We have a large amount of raw data that needs to be rewritten (via external API like GPT-4) in order to be useful. Current data selection methods only apply to rewritten data, which leads to wasted API calls for the samples that aren't selected (the rewriting for these samples becomes meaningless). Can the method proposed in this paper be extended to estimate the potential impact of each sample on the final task directly from the raw data, without rewriting? This would allow us to decide whether a sample is worth rewriting and selecting, thereby reducing unnecessary API calls.


It would be better if authors could provide some discussions and experiments.

**Questions For Authors:**

Please refer to the Strengths and Weaknesses.

**Relation To Broader Scientific Literature:**

Please refer to the Strengths and Weaknesses.

**Theoretical Claims:**

I'm not fully familiar with the concept and formulation of bandit. So I may miss some important details.

---

> ### Author Rebuttal · Authors · 2025-03-31
>
> Dear Reviewer xSam,
>
> Thank you very much for providing us with valuable feedback. We appreciate the detailed comments. Below, we have provided responses to each of your comments.
>
> ### Other Strengths And Weaknesses
>
> ***Can the method be applied to machine translation in ICL? How to select suitable samples to improve translation accuracy?***
> >  We would like to politely clarify that the primary objective of our work is to select exemplars *(input, rationale, output)* that effectively demonstrate the skills required to solve complex problems through reasoning over provided knowledge. In contrast, translation does not require stepwise reasoning using rationales and primarily consists of (input, output) pairs. While our bandit algorithm can be extended to select examples for translation tasks, this is *beyond the scope of our current work*. Adapting CASE to translation would **require modifications** to the **features** and the **format of LLM feedback (reward)** in the bandit-based selection algorithm. We **open-source our code to support reproducibility and facilitate extensions to new tasks**. To extend CASE to translation, BERTScore could be used as the metric for LLM feedback, and the feature could be based on the similarity between test instances and training instances in the source language.
>
> ***Can the method be extended to estimate the potential impact of each sample on the final task directly from the raw data, without rewriting?***
> > We are not clear on what **rewriting** exactly means here. However, to avoid the high costs associated with API calls, examples can be rewritten and selected using a smaller LLM and then transferred for inference on test instances using larger LLMs, as demonstrated in our work. Alternatively, by modifying the features and reward design (LLM feedback) in CASE, a subset of instances from raw data can be selected and prioritized for rewriting. We open-source our code to support adaptation to other data selection tasks. However, the task described in the query is **beyond the scope of our current work**, as our primary objective is to select *(input, rationale, output)* triplets by considering rationales as part of the selection process while capturing interactions between exemplars, unlike prior selection approaches.
>
>
>
> We would like to once again express our gratitude to Reviewer xSam for their valuable comments and suggestions. We will incorporate these insights into the revised manuscript. We believe our responses above effectively address all of Reviewer xSam's concerns. We will be happy to answer any further queries
>
> Sincerely,
>
> The Authors

---

### Official Review · Reviewer_FjhN · 2025-03-13

**Overall Recommendation:** 3

**Summary:**

This paper studies the in-context sampler selection problem using MAB.
The proposed method can work in isolation or combined with existing variants.
Results look promising.

**Claims And Evidence:**

I have questions regarding this part.
Can the authors elaborate more on the train-validation-test data split process?

**Essential References Not Discussed:**

I'm not aware of an important missing reference. But I'd like to see an experimental comparison over RAG and other zero-shot/multi-shot prompting methods (e.g., more ablation studies ranging from zero shot to 10 + shot).

**Experimental Designs Or Analyses:**

Can the author compare the time / compute of KNN and SC, as they also work well.

Table 1 reports the increase numbers using LENS as the baseline. However, according to the results, LENS is not the "second best" method, and seems to be significantly worse than EXPLORA. EXPLORA achieves similar performance as the proposed method.

In addition, have the authors considered the combination of EXPLORA with KNN / SC? It can help readers to better understand the trade-off between compute/LLM inference and performance.

**Methods And Evaluation Criteria:**

Besides the question above, I have a question regarding baselines:
Have the authors considered RAG as an additional baseline?

**Other Comments Or Suggestions:**

please see above sections.

**Other Strengths And Weaknesses:**

The idea is simple and makes sense.
The writing is clear, tables and figures look nice (though some of the figures and tables are designed to be misleading: e.g., figure 3 (a): there is no point of repeating several numbers as a figure; Figure 2 (c-d): error bars are missing?)

**Questions For Authors:**

please see details in the review sections above.

**Relation To Broader Scientific Literature:**

I'm not aware of important missing literature.

**Theoretical Claims:**

The application of sample-efficient MAB algorithms makes sense to me.

---

> ### Author Rebuttal · Authors · 2025-03-31
>
> Dear Reviewer FjhN,
>
> Thank you very much for providing us with valuable feedback. Below, we have provided responses to queries raised in the review.
>
> ***Train-Validation-Test data split***
> > The train-test split is provided in **Appendix C** and **Table 3**. For subset selection runs, we select $20$ validation examples. These 20 examples are chosen from a held-out set obtained by splitting the training set, rather than the original validation sets, to prevent data leakage. Here, we follow the same setup as EXPLORA for a fair comparison.
>
> ***Is RAG considered as an additional baseline?***
> > RAG entails retrieving context from external knowledge sources in an open-domain setting. Note that the **RAG and few-shot exemplar selection are complementary and not fundamentally competing approaches**. Also, the **tasks considered** in this work **are not well-suited for the RAG setting**. For instance, in the math word problem benchmarks like **GSM8K and AQUA**, all the information needed to solve a question is self-contained within the question itself. Similarly, benchmarks such as **FinQA and TabMWP** are intended to be evaluated in a reading comprehension setup where each question is closely tied to its context. Hence, they cannot be effectively evaluated in the open-domain setting of RAG.
>
> ***Time/Compute of KNN and SC***
> > Since CASE is a task-level exemplar selection method, it ensures that once exemplars are selected, they can be reused without incurring additional computational costs during inference. The time incurred by KNN and SC is during inference, and hence, **they are not comparable to exemplar selection time that occurs offline**.
>
> ***Table 1 reports increase numbers over LENS. However, it is not the "second best" method.***
> > The relative improvements of CASE over **EXPLORA** are shown in the Table below.
> > **Method**|GSM8K|AquaRat|TabMWP|FinQA|StrategyQA|
> > |---|---|---|---|---|---|
> > |**CASE**|79.91 (2.63%)|54.72 (2.20%)| 83.42 (0.04%)| 59.72 (0.43%)|84.49 (1.42%)|
> > |**CASE+KNN+SC**|87.49 (12.36%)| 64.17 (19.85%)| 86.23 (3.80%)|64.25 (8.05%)|85.92 (0.24%)|
> > |**CASE+MMR+SC**|85.60 (9.94%)| 62.60 (16.92%)| 85.91 (3.41%)|63.47 (6.74%)|84.69 (1.19%)|
>
> > We originally reported the gains over **LENS** because it was the **first task-level exemplar selection baseline**, making it a natural reference point. We acknowledge providing relative gains over EXPLORA and will incorporate them in the revised paper. The main advantage of CASE over EXPLORA is its **7×** efficiency gain (**Lines 411–413**) along with theoretical guarantees.
>
> ***EXPLORA with KNN/SC.***
> > We provide results for EXPLORA with KNN/SC in the table below. Due to space constraints, we were unable to include these in the original paper. We will incorporate these results in the revised paper.
> > **Method**|GSM8K|AquaRat|TabMWP|FinQA|StrategyQA|
> > |---|---|---|---|---|---|
> > |**EXPLORA**|77.86|53.54|83.07|59.46|85.71|
> > |**EXPLORA+KNN+SC**|85.89|64.17|85.74|63.64|86.53|
>
>
> ***Zero-shot/multi-shot prompting***
> > We employ 5-shot examples in our baselines to ensure a fair comparison. Below, we present zero-shot and multi-shot prompting. We observe that smaller LLMs are unable to fit more than 5 exemplars due to context length limits, and their performance plateaus beyond this point. We will incorporate these observations in the revised paper.
> > |**Method**| GSM8K| AquaRat|TabMWP|FinQA|StrategyQA|
> > |---|---|---|---|---|---|
> > |**Zero-shot**|67.02|38.15| 57.10|47.51|59.75|
> > |**1-shot**|67.55|38.58|66.3 |49.26|68.16|
> > |**3-shot**|68.99|41.33| 70.5|51.93 |70.00|
> > |**5-shot**|**73.46**|44.88|**71.22**|52.22|**73.06**|
> > |**7-shot**| 68.84| 44.88| 70.09|52.26|70.61|
>
> ***In figure 3 (a) numbers are repeated as figure***
> > We show that CASE makes **fewer LLM calls per iteration** than EXPLORA across benchmarks due to its optimized gap-index-based bandit approach. A table or text mentioning the relative difference might be effective, as the number of calls per iteration is similar across benchmarks, as it depends on the number of arms pulled per iteration.  Due to space constraints, we could not include a table in the original paper. In the revised paper, we will remove the figure and present the information in the text/table format instead.
>
> ***Figure 2 (c-d): error bars are missing?***
> > In Figures 2(c-d), we analyze the gap index and simple regret across rounds and observe that CASE converges similarly to existing bandit algorithms. These figures demonstrate that the principled approximation in CASE, ensures effective convergence. Since gap indices do not vary significantly across rounds, error bars were omitted for clarity of the plot. We will include this in the revised paper.
>
> Finally, we would like to thank Reviewer FjhN once again for these valuable comments. We will reflect these comments in the revised paper. We believe that our responses above address all of Reviewer FjhN's concerns and contribute to further strengthening our work.

---

> > ### Comment · Reviewer_FjhN · 2025-04-01
> >
> > Thank you for the additional results and response! I'm aware of the character limit on the author's responses, and I'd like to see some further explanation on the following points:
> >
> > - validation sample = 20: won't 20 be a too small number to draw statistically clear conclusions? Can the author reiterate the details of train-val-test split?
> >
> > - running time: I'm still curious about the running time, this is irrelevant to whether those are comparable but for a comprehensive understanding of the method.
> >
> > - can the authors summarize the message of those added experiments? EXPLORA + KNN + SC seems to have a strong performance.

---

> > > ### Author Response · Authors · 2025-04-01
> > >
> > > Dear Reviewer FjhN,
> > >
> > > We sincerely thank you for your positive feedback. Please find below our responses to the additional queries raised.
> > >
> > > ***1. Validation sample = 20 is small***
> > > > - We would like to clarify that the validation sample set of size 20 is used solely for sampling rewards from the LLM for selected arms during each iteration of the offline bandit selection runs. This set is not used for evaluation purposes. The sampled rewards are used to update the surrogate model within the bandit framework, which in turn assigns utility scores to exemplar subsets. For clarity, we will rename this to reflect reward sampling in the revised version of the paper.
> > > > - For evaluation, we use the **entire original test set**, and the results reported are based on the **original test sets** of the respective benchmarks. The details of these datasets are provided in **Section C of the Appendix**, and we also include statistical significance tests in **Table 1** of our submitted paper.
> > >
> > > ***2. Running time of KNN***
> > > > - The inference time of KNN is higher than that of using static selected exemplars from CASE. This is primarily due to the need for KNN to perform similarity searches over the entire training dataset by computing embedding similarities. For example, in the TabMWP benchmark, KNN is approximately **2× slower** than CASE per test instance during inference, as it must encode the test question and search through all 38,431 training samples.
> > > > - Inference time for KNN also varies across benchmarks, depending on the size of the training set. In general, KNN introduces **additional computational overhead during inference**, including the cost of input encoding and the similarity search over training exemplars, which are avoided with static selection methods like CASE. This distinction is also discussed in EXPLORA.
> > > > - However, hybrid strategies such as EXPLORA+KNN and CASE+KNN demonstrate that applying dynamic selection methods like KNN over a **reduced search space** can yield improved performance with less overhead during inference.
> > > > We report the average runtime per query (in seconds) for KNN and CASE below.
> > > > |**Method**        |  GSM8K    | AquaRat   | TabMWP  |
> > > > |---------------------|----------------|---------------|--------------|
> > > > | **KNN**    | 3.94$\pm$0.85 |    2.73$\pm$0.93     |    4.07$\pm$0.89    |
> > > > | **CASE**          |   2.40$\pm$0.35  |  1.77$\pm$0.55  |   1.69 $\pm$0.52      |
> > >
> > >
> > > ***3. EXPLORA + KNN + SC seems to have a strong performance***
> > > > We would like to highlight that CASE+KNN+SC outperforms EXPLORA+KNN+SC on three out of 5 benchmarks. Additionally, we would like to reiterate that the primary advantage of CASE over EXPLORA lies in its **7× efficiency gains** during offline selection and a significantly **reduced number of LLM calls**, as mentioned in **Section 4.2** of the submitted paper. Despite these efficiency gains, CASE remains competitive with, or marginally better than, EXPLORA in terms of task performance, as mentioned in **Lines 411-413** and **Lines 58-59**. Furthermore, CASE provides theoretical guarantees on sample complexity through its challenger set sampling strategy.
> > >
> > > We hope that we have addressed all the queries you raised, which improved the quality of our manuscript. If there are any remaining queries or additional clarifications needed, please let us know, and we would be happy to address them. Otherwise, we kindly request you to consider revising the score based on these updates.
> > >
> > > Sincerely,
> > >
> > > The Authors

---

### Official Review · Reviewer_j7VF · 2025-03-13

**Overall Recommendation:** 3

**Summary:**

The paper introduces a sample-efficient method for exemplar selection in ICL with LLMs. It formulates the selection of high-scoring exemplar sets as a top-$m$ best arms identification problem in stochastic linear bandits with a crafted linear reward model based on sentence similarity between exemplars and validation examples. Different with the existing GIFA algorithms, the method maintains a shortlist of challenger arms and selectively explores them, reducing the number of LLM evaluations required.

**Claims And Evidence:**

The main claims (computational efficiency, sample efficiency and performance improvement) are supported well by experimental results.

**Essential References Not Discussed:**

The summarized related works seems sufficient.

**Experimental Designs Or Analyses:**

The experimental design seems sound, and the results are sufficient. It would be better if the author could test the impact of exemplar subset size $k$ and validation set size $n'$ on performance.

**Methods And Evaluation Criteria:**

Benchmarks, GSM8K, FinQA, TabMWP, AquaRAT and StrategyQA, cover diverse reasoning tasks, and metrics (Exact Match, Cover-EM) are standard.

**Other Comments Or Suggestions:**

The introduction could be further improved to be more straightforward.

**Other Strengths And Weaknesses:**

This paper offers a new combination of ICL and MAB. The idea is interesting and convincing. The claims are supported well by the empirical results. The paper also provides good insights into the introduced method.

The reviewer's concern mainly lies in the validation sets. Since the selected exemplar sets are pre-fixed, how can them adapt to the tasks unseen in the validation tasks?

**Questions For Authors:**

Q1: How can the proposed method adapt to the tasks unseen in the validation tasks?

Q2: Do the performance gains hold for smaller LLMs?

Q3: What is the impact of exemplar subset size $k$ and validation set size $n'$ on performance?

---

Thanks for the rebuttal. I will keep my positive evaluation for this paper.

**Relation To Broader Scientific Literature:**

The work bridges ICL exemplar selection and top-$m$ bandit algorithms. It advances prior task-level methods [Rubin'22; Xiong'24; Ye'23] by integrating bandit-based exploration.

**Theoretical Claims:**

The proofs mainly follow (Reda et al., 2021). The correctness seems good.

---

> ### Author Rebuttal · Authors · 2025-03-31
>
> Dear Reviewer j7VF,
>
> Thank you very much for providing us with valuable feedback. We appreciate the detailed comments. Below, we have provided responses to queries raised in the review.
>
> ### Questions
>
> ***How can the proposed method adapt to the tasks unseen in the validation tasks?***
> > Our goal in this work is to select task-level exemplars that effectively demonstrate the skills needed to solve new instances related to a given task using the in-context learning ability of LLMs. CASE can be used to select task-level instances for any new task. However, **when tasks share one or more skills, exemplars selected for one task can be reused for another**, as LLMs learn to compose skills provided in in-context examples with already acquired knowledge [2, 3]. For example, FinQA requires table understanding, text understanding, and numerical reasoning skills, while TabMWP primarily focuses on table understanding and numerical reasoning. Thus, exemplars selected for the TabMWP task can be transferred to the FinQA task, as shown in [1], following task setups similar to those in [2, 3]. In the table below, we provide the results of using exemplars selected by CASE for TabMWP to solve FinQA. We observe that the model outperforms most state-of-the-art exemplar selection approaches that select exemplars from the FinQA training set. Additionally, its performance is close to that of CASE when selecting exemplars directly from FinQA, supporting the above-mentioned hypothesis.
> > |**Transfer from** |  **Target**   |     EM    |
> > |-----------------------|------------------|------------|
> > |   **TabMWP**    |  **FinQA**      |       55.36         |
>
> [1] In-Context Ability Transfer for Question Decomposition in Complex QA - Venktesh et. al. arXiv 2023.
>
> [2] Can Models Learn Skill Composition from Examples? - Zhao et. al. NeurIPS 2024.
>
> [3] Skill-Mix: A flexible and expandable family of evaluations for AI models - Yu et.al. ICLR 2024
>
>
> ***Do the performance gains hold for smaller LLMs?***
> > We have already reported the performance of exemplars from CASE on smaller models like **Mistral-7b** and **LLama2-7b** in the submitted manuscript. The results are shown in **Table 5**  and discussed in **Appendix D**. While emergent capabilities like in-context learning (ICL) and reasoning are more pronounced in large-scale models, we still observe that CASE achieves reasonable performance gains over other task-level/static exemplar selection methods across smaller open-source LLMs. Additionally, CASE remains competitive with instance-level/dynamic exemplar selection methods, further demonstrating its effectiveness. Its key advantage lies in **efficiency and reduced cost**, as it requires fewer LLM calls and optimization rounds due to the novel gap-index-based bandit algorithm.
>
> ***What is the impact of exemplar subset size k and validation set size n′ on performance?***
> > Thank you for your question. We adopt the values for exemplar subset size ($k$) and validation set size (n′) based on the values used in EXPLORA to ensure a fair comparison. Additionally, we also analyzed the impact of exemplar subset size ($k$) and validation set size (n′) on performance. Our findings show that increasing $k$ generally improves performance up to a certain point, beyond which additional exemplars provide diminishing returns or introduce noise, as shown in multi-shot prompting experiments in response to **reviewer FjhN**. Similarly, the choice of n′ impacts exemplar selection quality, as a sufficiently large validation set helps identify more representative exemplars, but excessive values can lead to overfitting or increased computational cost. We will clarify these observations in the revised manuscript.
>
>
> Finally, we would like to thank Reviewer j7VF once again for these valuable comments. We will reflect these comments in the revised manuscript. We believe that our responses above address all of Reviewer j7VF's concerns and contribute to further strengthening our work.
>
> Sincerely,
>
> The Authors

---

### Official Review · Reviewer_946f · 2025-03-14

**Overall Recommendation:** 3

**Summary:**

This paper investigates efficient example selection for ICL. It formulates the selection of exemplars as a top-m best arms identification problem. To address the challenge that the space of possible subsets (arms) is combinatorially large, the authors propose the sampling-based CASE method that maintains a shortlist of challenger arms and only pulls one of the arms from this shortlist. CASE results in a large reduction in LLM calls and running time. Theoretical analysis of the method and the experimental evaluations are presented.

**Claims And Evidence:**

The claims are well supported

**Essential References Not Discussed:**

N/A

**Experimental Designs Or Analyses:**

The experimental section is generally thorough, evaluating multiple baselines, two LLMs across five datasets, and includes synthetic experiments and ablation studies.

Potential issues include:
- There is a lack of introductions on how hyperparameters such as ϵ and Nt are chosen and ablation or sensitivity analysis regarding these hyper parameters.

**Methods And Evaluation Criteria:**

- The approach adopts an assumption that a reward for an arm can be modeled as a linear function of its features. However, in practice, it is likely nonlinear. Tasks with more complex interactions among examples may not fit this assumption, leading to suboptimal subset selection.

**Other Comments Or Suggestions:**

N/A

**Other Strengths And Weaknesses:**

**Additional Strengths**:
- The proposed method shows significant efficiency gains in LLM calls and example selection time.

- The transferability of exemples selected by smaller LLMs to larger LLMs enhances its practical usage

**Questions For Authors:**

N/A

**Relation To Broader Scientific Literature:**

The paper has discussed its difference from LENS and EXPLORA.

**Theoretical Claims:**

I did not examine the proofs for Lemma 1 in detail.

---

> ### Author Rebuttal · Authors · 2025-03-31
>
> Dear Reviewer 946f,
>
> Thank you very much for providing us with valuable feedback. We appreciate the detailed comments. Below, we have provided responses to queries raised in the review.
>
> ### Methods And Evaluation Criteria:
>
> ***Reward for an arm can be modeled as a linear function of its features. However, in practice, it is likely nonlinear.***
> > In this work, we focus on linear models for the following reasons:
> > - Recent works on top-k best arm selection [1] provide computationally simple and empirically tight bounds on the uncertainty of gap indices, $W_t(i, j)$,  for linear models.
> > - Small language models (e.g., Sentence-BERT) offer **high-quality pre-trained nonlinear feature maps** that can be effectively utilized with a linear model, ensuring both computational efficiency and empirically accurate confidence bounds.
> > - As described in **Lines 66-67** in the **Introduction Section**, recent works that develop a theoretical model for In-Context learning [2] primarily focus on linear functions. Their work demonstrates that trained transformers exhibiting in-context learning closely mimic familiar learning algorithms like ordinary least squares.  Hence, we employ a linear function based on sentence similarities between in-context examples and validation examples as a surrogate model for modeling the goodness of an in-context learning procedure.
>
> [1] Top-m identification for linear bandits - Reda et. al. AISTATS 2021.
>
> [2] Trained Transformers Learn Linear Models In-Context - Zhang et. al. JMLR 2024.
>
> ### Experimental Designs Or Analyses:
>
> ***How hyperparameters $\epsilon$ and $N_t$ are chosen and ablation or sensitivity analysis regarding them***
> > Thank you for your question. We have already introduced the hyperparameter selection process in **Section 4.1** of the submitted paper for both synthetic experiments and task-level exemplar selection experiments. For synthetic experiments, we **follow the standard setup** of prior bandit algorithms such as LinGIFA and LinGapE to ensure a **fair comparison**. The elements of $N_t$ are sampled as described in the algortihm. The size of $N_t$ is chosen such that $|N_t|<=|U_t|$, which is a sufficient condition to achieve convergence while bounding number of comparisons, as $N_t$ serves as the challenger set to the arms in $U_t$. This condition must be satisfied for convergence, and $|N_t|$ can be a user specified value satisfying the same. We repeated our synthetic experiments with various values for $|N_t|$ while ensuring this condition was met and observed that the algorithm converges in all cases. For real-world experiments, $|N_t|$  was fixed based on the evaluation on the **validation set**. Since $N_t$ varies in each iteration, it enables exploration of the space of arms while optimizing gap index computations per iteration. In summary, our findings show that while CASE is robust to small variations in these values, extremely large $\epsilon$ can lead to premature stopping, and significantly increasing $N_t$ may introduce unnecessary computation without notable performance gains. We will clarify these details further in the revised manuscript. We also have performed sensitivity analysis for different hyperparameters with results shown in **Figures 5 and 6** in the Appendix.
>
> We would like to express once again our gratitude to Reviewer 946f for their valuable comments and suggestions. We will incorporate these insights into the revised manuscript. We believe our responses above effectively address all of Reviewer 946f's concerns and further enhance the quality of our work.
>
> Sincerely,
>
> The Authors

---

### Decision · Program_Chairs · 2025-05-01

**Decision:**

Accept (poster)

**Comment:**

This paper addresses the challenge of efficiently selecting the most effective few-shot exemplars for ICL. The authors formulate this exemplar selection task as a top-m best arms identification problem within the stochastic linear bandits framework where each possible subset of exemplars represent an arm. They propose CASE, which maintains a dynamic shortlist of "challenger" arms (potential top candidates) and, in each iteration, selectively evaluates (pulls an arm via an LLM call) only one arm from either this shortlist or the current set of top-m arms.

After rebuttal, all reviewers have reached an unanimous decision of recommending acceptance of the paper -- they commended that 1) the main claims (efficiency, performance) are well-supported by experiments, 2) strong theoretical motivation of combining ICL and MAB, and 3) clarity of writing. Some issues were raised, such as the requests for more baselines and some hyperparameter details/analysis, but the authors were able to address these issues well during rebuttal. Overall, the AC concurs with the reviewers and recommend acceptance of this paper, but the AC also encourage the authors to consider the applicability and relative strengths of their method w.r.t modern advances of LLMs (e.g., long-context) which may reduce the need of demonstration selection.